# BAM-ICL: Causal Hijacking In-Context Learning with Budgeted Adversarial Manipulation

**Rui Chu[1]    Bingyin Zhao[2]    Hanling Jiang[1]    Shuchin Aeron[1]    Yingjie Lao[1]**

[1] Department of Electrical and Computer Engineering, Tufts University
[2] Meitu Inc

`{rui.chu, hanling.jiang, shuchin.aeron, yingjie.lao}@tufts.edu, bingyin@meitu.com`

## Abstract

Recent research shows that large language models (LLMs) are vulnerable to hijacking attacks under the scenario of in-context learning (ICL) where LLMs demonstrate impressive capabilities in performing tasks by conditioning on a sequence of in-context examples (ICEs) (i.e., prompts with task-specific input-output pairs). Adversaries can manipulate the provided ICEs to steer the model toward attacker-specified outputs, effectively "hijacking" the model's decision-making process. Unlike traditional adversarial attacks targeting single inputs, hijacking attacks in LLMs aim to subtly manipulate the initial few examples to influence the model's behavior across a range of subsequent inputs, which requires distributed and stealthy perturbations. However, existing approaches overlook how to effectively allocate the perturbation budget across ICEs. We argue that fixed budgets miss the potential of dynamic reallocation to improve attack success while maintaining high stealthiness and text quality. In this paper, we propose BAM-ICL, a novel budgeted adversarial manipulation hijacking attack framework for in-context learning. We also consider a more practical yet stringent scenario where ICEs arrive sequentially and only the current ICE can be perturbed. BAM-ICL mainly consists of two stages: In the offline stage, where we assume the adversary has access to data drawn from the same distribution as the target task, we develop a global gradient-based attack to learn optimal budget allocations across ICEs. In the online stage, where ICEs arrive sequentially, perturbations are generated progressively according to the learned budget profile. We evaluate BAM-ICL on diverse LLMs and datasets, the experimental results demonstrate that it achieves superior attack success rates and stealthiness and the adversarial ICEs are highly transferable to other models. Code is available at `https://github.com/CRcr0/BAM-ICL`.

## 1   Introduction

Recent development of large language models (LLMs) has revolutionized and empowered various fields, from reasoning Wei et al. [2022], Cheng et al. [2024a], Zhang et al. [2024] to math proof Azerbayev et al. [2023], Setlur et al. [2024], Didolkar et al. [2024] to protein design Madani et al. [2023, 2020], Cheng et al. [2024b], Ferruz and Höcker [2022]. Different from conventional models, LLMs also demonstrate remarkable capabilities in handling a wide range of problems and tasks through in-context learning (ICL) (a.k.a inference-time few-shot learning Brown et al. [2020], Garg et al. [2022], Xie et al. [2022], Min et al. [2022], Wies et al. [2023], Agarwal et al. [2024]). ICL is an intrinsic capability of LLMs that allows them to generate relevant responses to unseen input queries via "learning" from in-context examples (ICEs) (i.e., a sequence of prompts with task-specific input-output pairs), without updating model parameters. Although paving an effective path to undertake a variety of tasks by observing context examples, the potential risks and threats that the ICL ability may cause remain unclear and are worth exploring.

39th Conference on Neural Information Processing Systems (NeurIPS 2025).

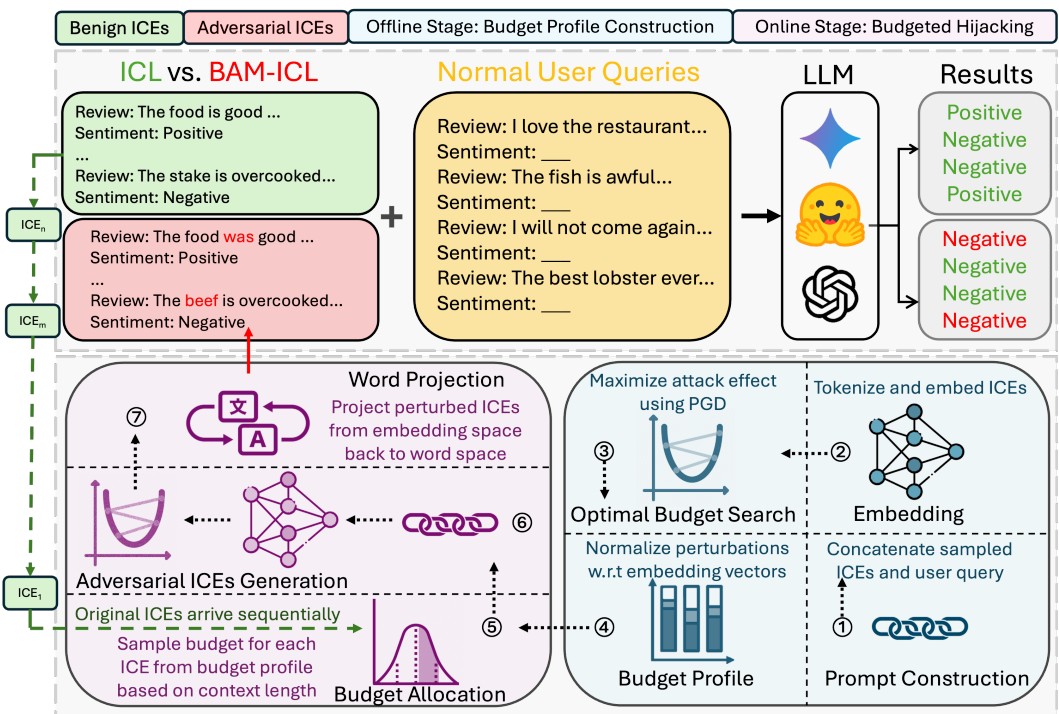

Figure 1: **Top: Illustration of ICL and BAM-ICL on LLMs. Bottom: Illustration of the framework design of BAM-ICL.** BAM-ICL hijacks LLMs and yields unintended output via adversarial ICEs (the block in red) while ICL with benign ICEs (the block in green) produces normal outputs. BAM-ICL is composed of two stages, where in the offline stage (the block in sky blue) we construct the budget profile to search for optimal budget distribution for each ICE and in the online stage (the block in light purple) we sequentially perform budgeted attack to generate adversarial ICEs.

In particular, it has been shown that the ICL capability of LLMs opens the door to model hijacking attacks Si et al. [2023], Qiang et al. [2023], Li et al. [2025], Salem et al. [2022], Jeong [2023], Kuo et al. [2025], where adversarial ICEs are used to steer victim models into producing attacker-specified outputs, effectively "hijacking" the model's decision-making process. A recent work Qiang et al. [2023] extends hijacking attack strategies from vision tasks to language models by using adversarial perturbations to craft malicious in-context examples. These adversarial example-based approaches modify input prompts to influence model behavior. In contrast to adversarial attacks Miyato et al. [2017], Wang et al. [2023a], Anwar et al. [2024], Li et al. [2020], Xhonneux et al. [2024], which have recently gained significant attention in the context of LLMs, hijacking attacks leverage the ICL mechanism to manipulate the model's behavior across a range of inputs, exhibiting distinct characteristics and challenges. 1) Instead of targeting a single input for misclassification, a hijacking attack perturbs the first few examples rather than a single example, with the objective of influencing the subsequent examples. To maintain stealthiness, the overall perturbation must be constrained. However, since the perturbation is distributed across multiple examples, the magnitude applied to each individual example can be reduced. 2) This also creates opportunities to dynamically allocate the perturbation budget among ICEs, which can further improve the attack performance (we show a fixed budget is suboptimal). 3) An adversarial attack is deemed successful when it identifies an adversarial example that misleads the model for a specific input; in contrast, a hijacking attack must maximize its influence on subsequent ICEs while operating under a predefined perturbation constraint. These observations motivate the central research question we seek to address: *How to design an effective and stealthy hijacking attack against LLMs, where subtle adversarial manipulations on ICEs gradually accumulate influence, analogous to a snowball effect, to steer the model's behavior?*

In this work, we propose BAM-ICL, a budgeted adversarial manipulation hijacking attack framework against LLMs through ICL. We consider a more practical yet stringent scenario where ICEs arrive sequentially and only the current ICE can be perturbed. As shown in Fig. 1, BAM-ICL mainly consists of two stages: In the offline stage, where we assume the adversary has access to data drawn

from the same distribution as the target task, we develop a global gradient-based budget profile construction algorithm to search for the optimal perturbation budget for each adversarial ICE given a total perturbation budget. In the online stage, where ICEs arrive sequentially, we progressively generate the perturbation for each ICE according to the budget profile constructed in the offline stage. To enhance the stealthiness and preserve the semantic meaning of ICEs, the perturbations are performed on the embedding space and then projected back to the word space. Our contributions are summarized as follows:

- To the best of our knowledge, this is the first work that exploits ICL to hijack LLMs with budgeted adversarial manipulation.
- We develop a novel two-stage attack framework composed of a global gradient-based attack that systematically searches for the optimal dynamic budget allocation strategy for individual ICEs and a refined causal attack that ensures the full utilization of the budget for each example.
- Through extensive experimentation on three benchmark datasets across various LLMs, we demonstrate that BAM-ICL achieves superior attack success rates and remarkable transferability while preserving the high text quality and stealthiness of adversarial ICEs.

## 2 Related Work

### 2.1 In-Context Learning

In-context learning (ICL) was first defined in Brown et al. [2020], where it was observed that pre-trained LLMs can perform new tasks by conditioning on a few task-specific demonstrations, without any parameter updates. This capability, often referred to as inference-time few-shot learning, enables the model to learn from a small set of input-output examples (ICEs) provided at test time. For example, an LLM is expected to predict the next token of {Prompt: Delicious fish! Output:___} given the following ICEs: {Prompt: I love this restaurant; Output: Positive. \n Prompt: The food is terrible; Output: Negative. \n}.

Early studies of ICL focused on the validation of hypotheses with synthetic experiments and provided insightful theoretical results Chan et al. [2022], Garg et al. [2022], Akyürek et al. [2023], Von Oswald et al. [2023], Hahn and Goyal [2023]. For instance, Agarwal et al. [2024] has demonstrated that many-shot in-context learning's superior learning capabilities on multiple datasets and benchmarks. Xie et al. [2022] proved that transformers and LSTMs are capable of inferring the hidden task-specific function in latent space from ICEs despite the mismatch between prompts and pretraining distributions using numerical synthetic data. Recent works have advanced progress and extended ICL to broader scopes Falck et al. [2024], Wang et al. [2023b]. Qiang et al. [2023] attempted to find good ICEs and generalized ICL from simple scenarios (e.g., number demonstrations) to complex real-world scenarios (e.g., natural language) and Agarwal et al. [2024] demonstrated that scaling the number of context examples leads to substantial performance improvements across a wide range of tasks. In this work, our objective is to explore the risks and threats ICL may bring to LLMs by such a new "learning" paradigm.

### 2.2 Attacks against Language Models

Threats against language models can be dated to attacks such as adversarial training methods Miyato et al. [2017] and HotFlip Ebrahimi et al. [2018]. There has been a line of works that leveraged adversarial attack Szegedy et al. [2014], a notorious inference-time attack against deep neural networks, and crafted adversarial examples to fool well-trained models by deliberately adding subtle adversarial perturbations on clean inputs using gradient-based approaches such as FGSM Goodfellow et al. [2015] and PGD Madry et al. [2018]. For example, FreeLB Zhu et al. [2020] perturbed the embedding layer the embedding layer of LLMs to manipulate the model output; TextFooler Jin et al. [2020] successfully deceived BERT Devlin et al. [2019] with adversarial examples (i.e., semantically similar alternatives that prioritize the most critical words to the model's prediction) and Ranjan et al. fooled GPT models in text classification by prioritizing influential tokens. However, these attacks do not exploit the in-context learning ability of LLMs. Recent security studies highlight that LLMs can be compromised via prompt-trigger attacks, and also propose unified defense mechanisms spanning prompt injection, backdoor, and adversarial prompts Lin et al. [2025]. Complementing

these prompt-level threats, their broader adversarial ML line covers imperceptible and arbitrary-target backdoors, ViT backdoor defenses, clean-label availability and class-oriented poisoning, and even neural-network operation backdoors or hardware-Trojan attacks Doan et al. [2022, 2023], Zhao and Lao [2022a,b], Clements and Lao [2019], Han et al. [2025], Hoang et al. [2024].

On the other hand, recent research has found that LLMs are vulnerable to hijacking attacks through in-context learning Ranjan et al., Kandpal et al. [2023], Zhao et al. [2024]. For instance, Li et al. [2025] reveals the essential factors of ICL robustness (e.g., model depth and context length) through context hijacking label manipulation. Anwar et al. [2024] investigates adversarial robustness in ICL for regression tasks, showing that perturbing input features (x-attacks) or context labels (y-attacks) can significantly degrade a transformer's ability to approximate the underlying function. Wang et al. [2023a] further shows that adversarial in-context examples transfer well across different models. A recent work (GGI) Qiang et al. [2023] proposed devising adversarial ICEs and hijacking LLMs by appending imperceptible malicious suffixes to in-context demonstrations using gradient-based optimization, and Kuo et al. [2025] explored the way to hijack against the safety reasoning mechanism. These works generally follow the design concept of conventional adversarial examples that impose the same perturbation on every single input. However, such a practice neglects the nature of ICL, in which models' predictions are steered by a sequence of demonstrations rather than a single example.

## 3 Method

### 3.1 Threat Model

The goal of hijacking attacks is to force a victim model $\mathcal{M}$ to generate manipulated output given new benign input queries via ICL on a sequence of adversarial ICEs. The attacker has no access to the internal parameters or training data of the model (i.e., black-box access), but can modify the ICEs that precede the actual input query. We consider a setting where the attacker knows the task (e.g., sentiment classification Socher et al. [2013], Zhang et al. [2015], Zampieri et al. [2019], Rosenthal et al. [2021]) and crafts adversarial ICEs $\{(x_i', y_i)\}_{i=1}^n$ with perturbed inputs but original labels to preserve stealth. The attack objective is thus to find a perturbed input set $\mathbf{X}' = \{x_1', x_2', \ldots, x_n'\}$. We assume that adversaries know the distribution of benign in-context examples and can sample instances for prompt manipulation, which is a reasonable assumption because normal context examples usually consist of simple input-output pairs that are relevant to the task and are consistent with prior works in related fields Zhao et al. [2021]. For example, in-context examples for the emotional classification task are often formulated as: $\{x :$ [Noun.] + [Linking verb] + [Adjective] (e.g., blast/dirty/fantastic); $y :$ [Pronoun] + [Linking verb] + [Adjective] (e.g., wonderful/negative/great)$\}$.

### 3.2 BAM-ICL

Given the threat model described above, the adversary's objective is to perturb a sequence of benign ICEs $\{(x_i, y_i)\}_{i=1}^n$ (where $y_i$ represents the ground-truth) to a set of adversarial examples $\{(x_i', y_i)\}_{i=1}^n$ such that, when presented to a language model $\mathcal{M}$ along with a new input query $x_{\text{query}}$, the model generates a manipulated output $\hat{y}_{\text{query}}$. We denote the set of adversarially perturbed inputs as $\mathbf{X}' = \{x_1', x_2', \ldots, x_n'\}$. Since the attack is conducted in the embedding space, we also denote their corresponding perturbed embeddings as $\mathbf{E}' = \{e_1', e_2', \ldots, e_n'\}$, where each $e_i'$ is derived by adding a constrained perturbation to the original embedding $e_i$ of $x_i$:

$$e_i' = e_i + \delta_i, \quad \text{s.t.} \quad \|\delta_i\| \leq \epsilon_i \tag{1}$$

The model's prediction conditioned on the ICE and the query input is given by:

$$\hat{y}_{\text{query}} = \mathcal{M}([x_1', y_1], \ldots, [x_n', y_n], x_{\text{query}}) \tag{2}$$

As aforementioned, LLMs exhibit the ability of "learning" to perform tasks simply by conditioning on a sequence of ICEs, which offers the opportunity to hijack the model's decision-making process through a series of adversarial ICEs rather than a single input. Therefore, we devise BAM-ICL based on this property of LLM. A general form of the BAM-ICL objective can be expressed as:

$$\max_{\mathbf{X}'} \mathcal{L}(\mathcal{M}(\mathbf{X}', x_{\text{query}}), y_{\text{query}}) \quad \text{s.t.} \quad d(x_i', x_i) \leq \epsilon_i \ \forall i \in \{1, \ldots, n\} \tag{3}$$

**Algorithm 1** Offline Phase: Budget Profile Construction

---

**Require:** Original ICE sequence $\mathbf{X}$, step size $\alpha$, number of steps $T$, total perturbation budget $\epsilon$

1: $\mathbf{P} \leftarrow \text{Prompt\_Construct}(\mathbf{X})$
2: $\mathbf{E} \leftarrow \text{Embedding}(\mathbf{P})$
3: Initialize $\mathbf{\Delta}^{(0)} \leftarrow \mathbf{0} \in \mathbb{R}^{\dim(\mathbf{E})}$
4: **for** $t = 0$ **to** $T-1$ **do**
5: $\quad \mathbf{\Delta}^{(t+1)} \leftarrow \text{Proj}_{\|\mathbf{\Delta}\|_2 \leq \epsilon}\Big(\mathbf{\Delta}^{(t)} + \alpha \, \nabla_{\mathbf{\Delta}_j} \mathcal{L}_{\mathbf{P}}^{(t)}\Big)$
6: **end for**
7: $\Gamma \leftarrow \text{Budget\_Profile}(\mathbf{\Delta})$
8: **return** $\Gamma$

---

where $d(\cdot)$ is the distance metric that measures the embeddings discrepancy between the original and perturbed inputs (i.e., $||e_i' - e_i||$). To ensure stealthy yet effective attacks, we set a total perturbation budget as a constraint for the ICEs:

$$\sum_{i=1}^{n} d(x_i', x_i) = ||e_i' - e_i||_2 \leq \epsilon \tag{4}$$

where $\epsilon$ is the overall perturbation budget. This constraint enables dynamic budget allocation across examples, which is the key design principle behind our BAM-ICL framework.

As shown in Fig. 1, the proposed BAM-ICL consists of two stages. In the offline stage (Section 3.3), we assume the adversary has access to data drawn from the same distribution as the target task, which allows simulating the ICL behavior of the target model. The goal of this stage is to construct an optimal perturbation budget profile for each adversarial ICE, under a total perturbation budget constraint. Specifically, we develop a global gradient-based optimization algorithm adapted from projected gradient descent (PGD) to determine the budget profile. We jointly optimize all perturbations under the shared constraint to calculate how the total budget $\epsilon$ should be distributed across individual examples. This optimization yields a dynamic budget allocation profile $\{\epsilon_i\}_{i=1}^{n}$, which encodes the relative sensitivity of each ICE. We construct a budget distribution $\Gamma$ based on $\{\epsilon_i\}_{i=1}^{n}$, which is stored and later used in the online stage to adaptively generate adversarial examples in real-time.

In the online stage (Section 3.4), the adversary no longer has access to future ICEs or queries in advance and must perturb each incoming ICE sequentially as it arrives. Since the number of ICEs $n$ can vary in practice, for each received input $x_i$, we retrieve the corresponding $\epsilon_i$ from the budget distribution $\Gamma$ based on $n$ and perform a constrained adversarial perturbation in the embedding space using a local PGD-based algorithm. The perturbed embedding $e_i'$ is then projected back to the word space to form the adversarial ICE $(x_i', y_i)$. This process continues until all $n$ ICEs have been processed. Crucially, the online stage only relies on the access to the current ICE and the precomputed budget profile, making it suitable for realistic settings where ICEs are streamed or constructed on the fly. By coupling global sensitivity insights from the offline phase with localized perturbations in the online phase, our framework ensures both efficiency and stealthiness in mounting hijacking attacks.

### 3.3 Offline Stage: Budget Profile Construction

In this stage, we aim to construct the budget profile. As presented in Algorithm 1, we design a PGD-based approach and run the algorithm offline where we assume the adversary has access to data drawn from the same distribution as the target task. Given the sampled ICE sequence, we first construct the prompt (Prompt\_Construct) by concatenating each ICE $x_i$ along with the corresponding ground-truth $y_i$ in order, followed by the user query $x_{\text{query}}$:

$$\mathbf{P} = ([x_1, y_1], \ldots, [x_n, y_n], x_{\text{query}}) \tag{5}$$

The prompt is then tokenized and embedded using a pre-trained embedding function: $\mathbf{E} = \text{Embedding}(\mathbf{P})$. Based on the dimension of the embedding $\mathbf{E}$, we initialize a perturbation vector $\Delta$. We then iteratively update the perturbation using gradient ascent on the loss function $\mathcal{L}_{\mathbf{P}} = \ell\left(M_\theta(\mathbf{P}_i(\Delta_i)), \mathbf{y}_{\text{Query}}\right)$ with respect to $\Delta$, subject to an overall $L_2$ constraint on the perturbation norm. At each step $t$, the perturbation is updated as expressed in line 5 of Algorithm 1, where $\alpha$ is the step size and $\text{Proj}_{\|\Delta\|_2 \leq \epsilon}(\cdot)$ denotes the projection onto the $L_2$ ball with radius $\epsilon$ to ensure that the total perturbation budget is not exceeded.

---

**Algorithm 2** Online Phase: Budgeted Hijacking Attack

---

**Require:** original ICE sequence $\mathbf{X} = \{x_1, x_2, ..., x_n\}$, step size $\alpha$, number of steps $T$, budget profile $\Gamma$, total perturbation budget $\epsilon$, context length $n$

1: $\{\gamma_1, \gamma_2, ..., \gamma_n\} \leftarrow \text{Calc\_Budget}(\Gamma, n)$
2: $\mathbf{P} \leftarrow [\,]$
3: **for** $i = 1$ **to** $n$ **do**
4:     $\mathbf{P} \leftarrow \mathbf{P} + \text{Prompt\_Construct}(x_i)$
5:     $e_i = \text{Embedding}(\mathbf{x_i})$
6:     Initialize $\delta_i^{(0)} \leftarrow \mathbf{0} \in \mathbb{R}^{\dim(e_i)}$
7:     $\epsilon_i = \gamma_i \cdot \epsilon$
8:     **for** $t = 0$ **to** $T - 1$ **do**
9:         $\delta_i^{(t+1)} \leftarrow \text{Proj}_{\|\delta_i\|_2/\|e_i\|_2 \leq \epsilon_i}\left(\delta_i^{(t)} + \alpha \nabla_{\delta_i} \mathcal{L}_{\mathbf{P}}^{(t)}\right)$
10:     **end for**
11:     $x_i' \leftarrow \text{Word\_Proj}(e_i, \delta_i^T, \epsilon_i)$
12: **end for**
13: **return** $\mathbf{X}' = \{x_1', x_2', ..., x_n'\}$

---

After $T$ iterations, we obtain the perturbed embedding:

$$\mathbf{E}' = \big([e_1 + \delta_1, \ \text{Embedding}(y_1)], \ \ldots, \ [e_n + \delta_n, \ \text{Embedding}(y_n)], \ e_{\text{query}}\big) \qquad (6)$$

where $e_i = \text{Embedding}(x_i)$ and $\delta_i$ is the embedding perturbations for each $e_i$. Thus, we have $\Delta_n = (\delta_1, \ldots, \delta_n)$. We then compute the relative perturbation magnitude allocated to each example by normalizing the $L_2$ norm of each perturbation with respect to the corresponding embedding vector:

$$\gamma_i = \frac{\|\delta_i\|_2/\|e_i\|_2}{\sum_{j=1}^n \|\delta_j\|_2/\|e_j\|_2} \qquad (7)$$

The resulting set $\{\gamma_i\}_{i=1}^n$ forms the budget profile $\Gamma$. This dynamic allocation captures the relative sensitivity or influence of each ICE on the model's behavior and is stored for use in the online phase.

### 3.4 Online Stage: Budgeted Hijacking Attack

In the second stage of BAM-ICL, we perform the budgeted hijacking attack online where ICEs arrive sequentially. The details of our method are presented in Algorithm 2. We first sample the corresponding budget for each ICE according to the context length $n$ from the budget profile $\Gamma$ that is constructed in the offline stage as discussed above.

For each incoming ICE $x_i$, we follow a similar process as in the offline stage to construct the prompt (Prompt\_Construct) and generate the embedding (Embedding) (i.e., lines 4 and 5 in Algorithm 2). Please note that, for ICE $\mathbf{x_i}$, unlike in the offline stage, we retain all previously perturbed inputs and hence embeddings.

Using the sampled budget $\gamma_i$, we scale the perturbation budget for each ICE, setting $\epsilon_i = \gamma_i \cdot \epsilon$, where $\epsilon$ is the total perturbation budget. We then employ a PGD-based optimization to progressively generate the perturbation by computing the gradient of the loss function $\mathcal{L}_{\mathbf{P}}$ with respect to the perturbation $\delta_i$ and updating the perturbation vector using the step size $\alpha$ (i.e., lines 6 to 10 in the algorithm). This iterative process continues for $T$ steps, gradually adjusting the perturbation within the $L_2$ constraint. Finally, the perturbed ICE $x_i'$ is obtained by projecting the perturbed embedding $e_i$ back into the word space by using Algorithm 3, resulting in the hijacked ICE sequence $\mathbf{X}' = \{x_1', x_2', ..., x_n'\}$.

Algorithm 3 is designed to project perturbed embeddings back into the word space by finding the semantically closest words to the perturbed embedding $e_i + \delta_i$. This ensures that the perturbation is applied in a way that maintains semantic coherence while making it difficult to detect. The first step of the algorithm constructs a candidate set $\mathcal{C}$, which consists of words from the dictionary $\mathcal{D}$ whose embeddings are within the perturbation boundary. Specifically, the algorithm selects words $w$ from the dictionary where the $L_2$ distance between the embedding of $w$ and the original embedding $e_i$ is within the perturbation boundary $\epsilon_i$. This ensures that only words with embeddings that are semantically close to $e_i$ are considered as potential candidates for the projection. Next, from this

**Algorithm 3** Word_Proj()    Word Projection Back from Embedding Space

---
**Require:** embedded context $e_i$, perturbation $\delta_i$, perturbation budget $\epsilon_i$, dictionary $\mathcal{D}$, top-$k$ value $k$
  1: $\mathcal{C} \leftarrow \{ w \in \mathcal{D} \mid \|\text{Embedding}(w) - e_i\|_2 \leq \epsilon_i \}$
  2: $\mathcal{K} \leftarrow \arg\min_{w \in \mathcal{C}}^{k} \|\text{Embedding}(w) - (e_i + \delta_i)\|_2$
  3: $x_i' \leftarrow \arg\max_{w \in \mathcal{K}} \mathcal{L}\big(\text{Replace}(x_i, w)\big)$
  4: **return** $x_i'$

---

candidate set $\mathcal{C}$, the algorithm selects the top-$k$ words that are closest to the perturbed embedding $e_i + \delta_i$. Then, we choose the word $x_i'$ from this set that maximizes the model's loss function. This step ensures that the selected word not only remains semantically similar to the original word but also optimizes the effectiveness of the attack by influencing the model in the intended direction. It is worth noting that the choice of the word projection function is flexible and can be substituted with other methods Guo et al. [2024], Gonen et al. [2023], Stolfo et al. [2025].

## 4 Experiments

### 4.1 Experimental Settings

**Datasets and models.** We follow the same practice in existing attacks Jeong [2023] against LLMs and evaluate BAM-ICL on SST-2 Socher et al. [2013], AG's News Zhang et al. [2015] and OLID Rosenthal et al. [2021]. These datasets are common text-classification benchmarks that cover a wide range of tasks, including sentiment analysis, topic categorization, and offensive language detection. For victim models, we select various model families that span from 1B to 30B, including GPT2-XL Radford et al. [2019], LLaMA Touvron et al. [2023], OPT Zhang et al. [2022], and Mistral Jiang et al. [2023]. Concrete details on the models and datasets are summarized in the appendix. All experiments are performed on NVIDIA L40S GPUs.

**Attack configurations and baselines.** We adopt the prompt construction and guiding sentence strategy from Qiang et al. [2023], combined with the sequential masking logic introduced by Garg et al. [2022]. For each ICE, we allow up to three tokens to be modified. The context length $n$ ranges from 2 to 12, consistent with prior works Qiang et al. [2023], Kandpal et al. [2023], Li et al. [2024]. We evaluate our method against two baselines: a flat budget allocation strategy and the GGI attack Qiang et al. [2023], which performs global perturbation across all ICEs.

**Metrics.** We comprehensively evaluate the generation performance, attack effectiveness, ICEs text quality, and stealthiness in our experiments. We report *Clean Accuracy (CA)* to show the normal generation performance of the language models. For the attack effect, we employ the standard *Attack Success Rate (ASR)* (i.e., the percentage of manipulated ICEs that successfully yield malicious behavior) as the criteria. A higher ASR indicates better attack performance. We adopt *Perplexity* Bahl et al. [1983] and *Cosine Similarity* to evaluate the stealthiness and text quality of adversarial ICEs, respectively. ICEs with high stealthiness and text quality tend to have cosine similarity values close to 1 (i.e., more similar to benign ICEs) and low perplexity values. We also compute the ASR drop against defenses as an orthogonal assessment for stealthiness, a stealthier attack can evade the defense and maintain the ASR (i.e., lower ASR drop).

### 4.2 Attack Effectiveness

We first demonstrate the attack effectiveness of BAM-ICL. We perform the hijacking attack with BAM-ICL and the baseline methods including the hijacking attack with a flat budget (i.e., attack with equally allocated perturbation towards each ICE) and global hijacking attack (i.e., attacking all ICEs simultaneously, the first stage of BAM-ICL). The results on various OPT models are shown in Table 1. It can be seen that compared to the flat budget attack where each ICE receives an even perturbation budget, BAM-ICL achieves superior ASR on all datasets across various models with two different context lengths, indicating the effectiveness of the design with dynamic budget allocation. The results for other models are reported in the appendix, which show similar trends.

Comparison to prior hijacking attack GGI Qiang et al. [2023] is shown in Fig. 2. It is important to note that GGI is also a global attack that perturbs all the ICEs simultaneously. Thus, it is understandable

that GGI and the baseline global attack achieve slightly better ASR. However, unlike GGI that adds an easy-to-detect suffix and compromises the semantic meaning of the sentence, BAM-ICL preserves the quality of the manipulated ICEs with the design of the word project function, which also improves the stealthiness of the attack. We measure the perplexity score on 100 randomly sampled perturbed ICEs as shown in Fig. 2(b). It can be seen that the perplexity score of the manipulated ICEs under our BAM-ICL remains similar to the clean ICEs, ensuring low perceptibility.

Table 1: Attack Sucess Rate (ASR) on different OPT models

| Method | OPT-1.3B | | | OPT-6.7B | | | OPT-13B | | | OPT-30B | | |
|---|---|---|---|---|---|---|---|---|---|---|---|---|
| | SST-2 | AGNews | OLID | SST-2 | AGNews | OLID | SST-2 | AGNews | OLID | SST-2 | AGNews | OLID |
| | | | | | | $n=3$ | | | | | | |
| CA | 86.85 | 68.60 | 70.14 | 89.16 | 73.20 | 71.08 | 90.04 | 72.90 | 71.54 | 92.45 | 71.00 | 74.69 |
| +Global | $70.18_{\pm 2.15}$ | $39.64_{\pm 1.83}$ | $53.21_{\pm 3.45}$ | $64.55_{\pm 2.06}$ | $35.70_{\pm 3.17}$ | $48.95_{\pm 2.18}$ | $60.79_{\pm 3.25}$ | $34.11_{\pm 2.59}$ | $46.33_{\pm 2.96}$ | $57.46_{\pm 3.05}$ | $32.88_{\pm 1.76}$ | $44.02_{\pm 3.44}$ |
| +Flat | $37.92_{\pm 2.01}$ | $17.39_{\pm 1.86}$ | $26.54_{\pm 1.63}$ | $33.49_{\pm 2.58}$ | $15.36_{\pm 2.48}$ | $23.77_{\pm 3.09}$ | $31.12_{\pm 1.95}$ | $14.87_{\pm 1.87}$ | $22.66_{\pm 3.22}$ | $29.34_{\pm 2.91}$ | $14.06_{\pm 2.48}$ | $21.51_{\pm 2.03}$ |
| +BAM-ICL | $47.32_{\pm 3.96}$ | $24.87_{\pm 2.18}$ | $35.66_{\pm 3.31}$ | $41.99_{\pm 2.94}$ | $22.51_{\pm 2.36}$ | $32.49_{\pm 3.61}$ | $39.14_{\pm 3.54}$ | $21.36_{\pm 2.15}$ | $30.72_{\pm 3.15}$ | $36.85_{\pm 3.38}$ | $20.21_{\pm 3.01}$ | $29.44_{\pm 2.41}$ |
| | | | | | | $n=12$ | | | | | | |
| CA | 88.85 | 70.60 | 72.14 | 91.16 | 75.20 | 73.08 | 92.04 | 74.90 | 73.54 | 94.45 | 73.00 | 76.69 |
| +Global | $74.66_{\pm 2.74}$ | $43.25_{\pm 1.93}$ | $57.41_{\pm 2.65}$ | $68.55_{\pm 3.02}$ | $40.36_{\pm 3.15}$ | $52.66_{\pm 2.38}$ | $65.17_{\pm 2.81}$ | $38.84_{\pm 1.64}$ | $50.20_{\pm 3.44}$ | $62.33_{\pm 3.81}$ | $37.45_{\pm 1.99}$ | $48.14_{\pm 3.07}$ |
| +Flat | $41.55_{\pm 2.11}$ | $22.74_{\pm 1.59}$ | $31.61_{\pm 2.55}$ | $36.26_{\pm 2.44}$ | $20.45_{\pm 2.79}$ | $28.30_{\pm 3.08}$ | $34.14_{\pm 2.28}$ | $19.62_{\pm 2.44}$ | $26.79_{\pm 2.79}$ | $31.89_{\pm 2.65}$ | $18.90_{\pm 2.01}$ | $25.40_{\pm 2.75}$ |
| +BAM-ICL | $52.71_{\pm 2.66}$ | $30.66_{\pm 2.23}$ | $40.98_{\pm 3.52}$ | $47.55_{\pm 2.84}$ | $27.80_{\pm 3.35}$ | $36.55_{\pm 2.59}$ | $44.78_{\pm 2.76}$ | $26.45_{\pm 1.98}$ | $34.92_{\pm 3.37}$ | $42.14_{\pm 3.02}$ | $25.30_{\pm 2.51}$ | $33.49_{\pm 2.11}$ |

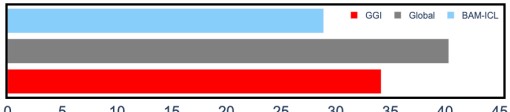

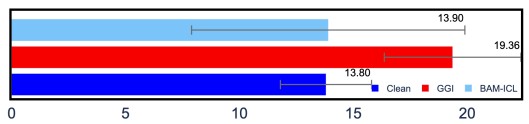

(a) Averaged ASR comparison on AGNews with a context length of 4.

(b) Averaged perplexity score from 100 randomly sampled perturbed ICEs.

Figure 2: Comparison of ASR and text quality between BAM-ICLand baseline attacks.

We also visualize the budget profile under runs with different total budgets, i.e., $\epsilon$, as shown in Fig. 3, which reveals the importance of the budget profile construction in the offline stage.

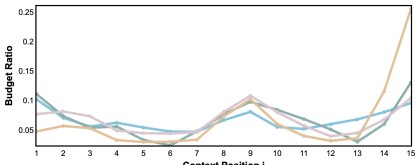

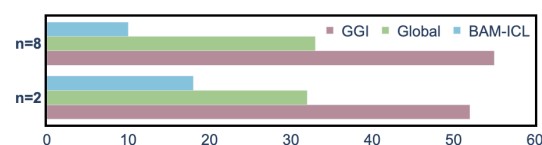

Figure 3: Budget profile on SST-2.

Figure 4: ASR drop ($\Delta$ASR) against defense on SST-2.

## 4.3 Performance against Defense

To better understand the stealthiness of BAM-ICL, we also evaluate the performance against the defense proposed in Qiang et al. [2023], which prepends clean ICEs of the same context length $n$ to the manipulated ICE sequences. Fig. 4 shows the ASR change between before and after prepending clean ICEs under the hijacking attack. BAM-ICL demonstrates superior performance in evading existing defense strategies, especially with increased context lengths, achieving substantially lower ASR drop against the defense compared to both global attack and the attack in Qiang et al. [2023]. BAM-ICL demonstrates stronger resilience against filtering Jain et al. [2023] and detection-based Nguyen and Wong [2023] defenses than prior work Qiang et al. [2023], both within individual ICEs (ASR drop by defense for 19.63 compared to prior work at 39.83) and across multiple ICEs (19.66 compared to 22.32). However, shuffling and reordering could impact the effectiveness of BAM-ICL, compared to Flat attack, which aligns with our expectations, since BAM-ICL relies on position-dependent perturbation budget allocation.

## 4.4 Stealthiness

Another notable advantage of BAM-ICL is the stealthiness. Ideally, a stealthy enough attack can generate cosmetically and semantically similar adversarial ICEs to the original benign text. Fig. 5 provides insights into the cosine similarity distributions, which measure semantic alignment between adversarial ICEs and the original text. BAM-ICL consistently exhibits higher similarity (i.e., close to 1), indicating a better semantic preservation after attack compared to the flat budget counterpart.

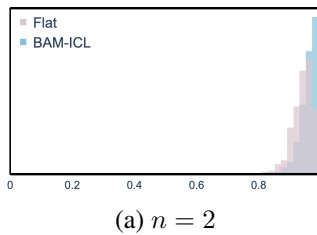 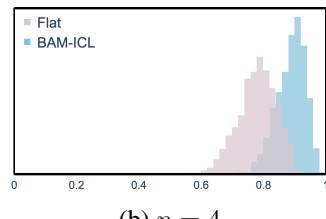 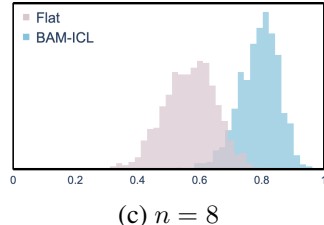

(a) $n = 2$          (b) $n = 4$          (c) $n = 8$

Figure 5: Cosine similarity distribution histograms of ICEs under various context lengths.

## 4.5 Transferability

We then demonstrate the transferability of adversarial ICEs generated by BAM-ICLby applying them to LLMs other than the original victim model. Using adversarial ICEs crafted from the SST-2 task on the OPT model, we query the *DeepSeek-Chat* API (based on the V3 model) with the same inputs, which shows a similar ASR across models. More importantly, we find that not only does adversarial ICE exhibit high transferability, but also the learned budget profile generalizes well across different contexts and model configurations. We apply the budget profile learned from the SST-2 task on the OPT model with a context length of 12 to GPT2-XL and report the results in Table 2.

Table 2: ASR for budget profile transferability from SST-2

| Modified Tokens | $n = 4$ | | | $n = 8$ | | |
|:---:|:---:|:---:|:---:|:---:|:---:|:---:|
| | OPT | $\longrightarrow$ | GPT-XL | OPT | $\longrightarrow$ | GPT-XL |
| 1 | 0.61 | | 0.51 | 0.78 | | 0.73 |
| 2 | 0.75 | | 0.63 | 0.89 | | 0.86 |
| 3 | 0.93 | | 0.93 | 0.99 | | 0.97 |

## 4.6 Time Complexity

For each run of the offline phase, we select input–output pairs equal in number to the attack context length from the training set. The budget profile is averaged over multiple runs. During the online phase, the full test set is used for evaluation. All results are normalized by the time required to compute perturbations per ICE using the Global attack. It can be seen that BAM-ICL offers much lower time complexity than the Global attack; even accounting for the additional cost of the offline phase, the overall runtime increase remains modest.

Table 3: Average runtime for time complexity comparison

| Method | Offline Total | Offline /ICE | Online /ICE | Overall/ICE |
|:---|:---:|:---:|:---:|:---:|
| Global | — | — | 1.00 | 1.00 |
| Flat | — | — | 0.42 | 0.42 |
| BAM-ICL | 13.60 | 0.68 | 0.41 | 1.09 |

## 4.7 ICL on Linear Functions

Besides LLM evaluation benchmarks, we also evaluate the performance on a linear task, which is well-studied with theoretical foundations for ICL Garg et al. [2022], Xie et al. [2022], Anwar et al. [2024], Li et al. [2023]. Following Garg et al. [2022], we first trained a small transformer model on numerical linear functions, which has been shown to be capable of performing a specific learning

function entirely via inference from ICEs (more details are presented in the appendix). We then perform both attacks with a flat budget and BAM-ICL. As shown in Fig. 6, the transformer trained in this numerical setting exhibits behavior similar to that of LLMs. When applying a learned budget profile (Fig. 6(a)), the loss increases more rapidly than with the attack with a flat budget (Fig. 6(b)). These results show that BAM-ICL is well generalizable.

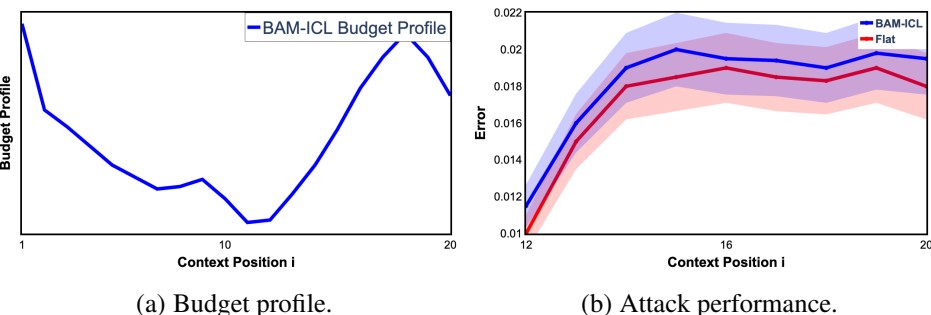

(a) Budget profile.

(b) Attack performance.

Figure 6: Results on linear functions.

## 4.8 Generalizability to Other Tasks

The hijacking attack manipulates an output of LLM through ICL to deliberately steer its intended behavior. In contrast, jailbreaking Wei et al. [2023] focuses on bypassing a model's safety guardrails (e.g., human alignment) to override ethical or safety constraints, while the backdoor attack Kandpal et al. [2023] implants malicious behaviors via crafted demonstrations and triggers them using prompts containing predefined triggers.

BAM-ICL can also generalize to other attack scenarios, such as jailbreaking tasks, demonstrating its broader applicability. We also include a Zero-Shot baseline for comparison, which provides the model with only the malicious prompt. As shown in Table 4, the overall performance of all methods degrades under this scenario. To improve attack effectiveness, one viable approach is to increase the context length of $n$. Consistent with our earlier findings, our method continues to benefit from larger $n$ and consistently outperforms prior work, further validating the effectiveness of the proposed budgeted strategy.

Table 4: Jailbreaking Results on LLaMA-3.1

| Method | $n = 2$ | $n = 4$ | $n = 12$ |
|---|---|---|---|
| Zero-Shot | 2.2 | 2.2 | 2.2 |
| GGI | 10.6 | 24.4 | 31.7 |
| BAM-ICL | 7.9 | 17.4 | 43.6 |

## 5   Conclusion

In this work, we propose BAM-ICL, a novel budgeted hijacking attack framework against LLMs under ICL. Unlike conventional adversarial methods, BAM-ICL strategically allocates perturbation budgets across ICEs to maximize influence while maintaining stealthiness. Our two-stage design first learns a global budget profile offline using a gradient-based optimization method, then applies online perturbations to ICEs as they arrive sequentially. Experimental results across diverse LLMs and tasks confirm the effectiveness, transferability, and stealthiness of our approach. Overall, BAM-ICL demonstrates that dynamic, budget-aware adversarial manipulation poses a serious and practical threat to LLMs operating under ICL.

## Acknowledgment

Rui Chu and Yingjie Lao are supported in part by the National Science Foundation (NSF) SaTC-2426299 and SaTC-2413046. Shuchin Aeron and Hanling Jiang would like to acknowledge support by NSF GCR: 2428640.

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

# NeurIPS Paper Checklist


## A    Summary of Appendix

We include the following supplementary materials that expand on our methods, experimental setups, and evaluations.

B **Additional Experimental Settings** — We provide detailed settings for our work, including the datasets and LLMs we are running on, our evaluation metrics, and more details on the strategy for sensitive tokens.

C **Additional Experiments** — We provide a detailed comparison of different models (OPT family and LLaMA family, as well as Mistral) with different datasets and different context lengths, to show the effectiveness of our methods under different $\epsilon$ and different *Modified Token* amounts. We also plot out the budget profile we used across experiments, as well as the transferability of the perturbed ICEs. Results of generalized tasks are also provided.

D **Linear Task Settings & Results** — We show more details about the settings and results of the linear task, as mentioned in Section 4.7 of the main paper.

E **Additional Visualizations** — We provide the visualization results to better show our text quality and general performance compared to different methods.

F **Limitations** — We discussed the limitations of our works.

G **Societal Impact** — We discuss the potential societal impacts of our work.

H **Prompt Examples** — We show clean and perturbed examples.

## B    Additional Experimental Details

### B.1    Datasets, LLMs, and Metrics

#### B.1.1    Datasets

- **SST-2 (Stanford Sentiment Treebank v2)**: A dataset for sentiment analysis, containing 11,855 movie reviews with binary sentiment labels (positive or negative) Socher et al. [2013].
- **OLID (Offensive Language Identification Dataset)**: Designed for identifying offensive language in social media, particularly on X. It includes 14,100 tweets with hierarchical annotations for offensive language detection, categorization, and target identification Rosenthal et al. [2021].
- **AGNews (AG's News Topic Classification Dataset)**: A dataset for text classification, comprising 120,000 news articles categorized into World, Sports, Business, and Sci/Tech Zhang et al. [2015].

#### B.1.2    LLMs

- **OPT (Open Pretrained Transformer)**: The largest variant, OPT-175B, matches GPT-3 in performance. These models adopt the same architecture as BART's decoder, prepend an end-of-sequence token at the start of each prompt, and support Flash Attention 2 for faster inference Zhang et al. [2022]. In our experiments, we experimented on OPT family from 1.3 B to 30B.
- **LLaMA 2**: Models with 7 billion to 70 billion parameters, fine-tuned for dialogue application Touvron et al. [2023]. Trained on 2 trillion tokens with a 4096-token context window.
- **LLaMA 3 series**: A specialized branch of the LLaMA family, LLaMA 3.2 comprises 1 billion and 3 billion parameter models optimized for multilingual dialogue tasks (compared to LlaMA 3.1-8b-Instruct). Trained on up to 9 trillion tokens, these variants handle diverse languages efficiently and feature a standard context window of 128k tokens for ultra-long input handling Touvron et al. [2023].
- **Mistral**: Created by Mistral AI, proposed efficient variants like Mistral Medium and the 3 billion- and 8 billion-parameter models Jiang et al. [2023].
- **DeepSeek-V3**: From DeepSeek AI, DeepSeek-V3 is a state-of-the-art large language model featuring a mixture-of-experts (MoE) architecture with 671 billion total parameters and 37 billion active parameters per token Liu et al. [2024]. It is open-sourced for researchers.

### B.1.3 Metrics

**Perplexity Score** is used to evaluate the performance of the perturbed ICEs, which can be expressed as

$$\text{PPL} = \exp\left(-\frac{1}{A}\sum_{g=1}^{A}\log p(w_g \mid w_{<g})\right) \tag{8}$$

where $A$ is the total number of tokens in the sequence, $g$ is the index of the $g$-th token, ranging from 1 to $A$, $w_{<g} = \{w_1, w_2, \ldots, w_{g-1}\}$ is the preceding context of length $g-1$, and $p(w_g \mid w_{<g})$ is the conditional probability assigned by the language model to token $w_g$ given its prior context.

**Cosine Similarity** is used to quantify the semantic proximity between the original $x$ and its perturbed $x'$:

$$\text{cosine\_similarity}(x', x) = \frac{{x'}^{\top} x}{\|x'\|_2 \|x\|_2}. \tag{9}$$

where

$$\|x\|_2 = \sqrt{x^{\top} x}.$$

which is the $l_2$ norm of $x$. Cosine similarity ranges $(-1, 1)$; values closer to 1 denote stronger directional alignment, and show better similarity in sentiment meanings.

**Loss** in our implementation can be given by

$$\mathbf{h}_g = \big(\text{Transformer}(\text{Embedding}[w_{1:g}])\big)_g, \tag{10}$$

$$\Pr(y_{g+1} \mid \mathbf{h}_g) = \text{softmax}(\mathbf{z}_{g+1})_{y_{g+1}}, \tag{11}$$

$$\ell_{g+1} = -\log \Pr(y_{g+1} \mid \mathbf{h}_g). \tag{12}$$

where $g \in \{1, \ldots, A\}$ is the index position of the current input token within a sequence of length $A$, $\mathbf{h}_g \in \mathbb{R}^d$ is the hidden state at $g$, $z_{g+1}$ is the pre-softmax logit assigned to candidate token $w$ when predicting position $g+1$.

## B.2 Hyperparameter Selections

To automate hyperparameter selection for the perturbation generation, we treat both the step size $\alpha$ and times $t$ as variables in an optimization problem. Optuna's Tree-structured Parzen Estimator (TPE) Akiba et al. [2019] sampler iteratively proposes candidate pairs and receives feedback via an objective that reflects adversarial strength.

## B.3 Sensitive Token Selection

**Tokenization** Assume the selected ICE $x_i$ contains $A$ tokens. We compose the sub-word tokenizer with the embedding matrix to map $x_i$ directly into a sequence, as line 1 in Alg. 4 , where the tokenizer (e.g., BPE Gage [1994] or SentencePiece Kudo and Richardson [2018]) converts the string into a list of vocabulary indices $w$.

**Input vector construction.** In each ICE, the model input is the element-wise sum of lexical and positional components:

$$w = e + g \tag{13}$$

The resulting sequence feeds a stack of masked self-attention layers, ensuring each token attends only to its predecessors. $g$ is the positional encoding for the token $w$. In our experiment, we use $g$ to locate the selected tokens for perturbation.

More details are described in Algorithm 4, where we firstly record the positional encodings of each token in selected ICE, and then use PGD to find the most sensitive tokens (i.e., lines 3 to 11 in Algorithm 4). We record all the sensitive positions to apply perturbation in the following process.

**Algorithm 4** PGD-Based Sensitive Position Encoding Selection

---

**Require:** selected ICE $x_i$, label $y$, step size $\alpha$, steps $T$, budget $\epsilon$, top-$m$ selected tokens $m$, total token amount in this ICE $A$
1: $(w_1, \ldots, w_A) \leftarrow \text{Tokenizer}(x_i)$
2: $g \leftarrow \text{PositionalEncoding}(w)$
3: **for** $g = 1$ **to** $A$ **do**
4:     $\boldsymbol{\delta}^{(0)} \leftarrow \mathbf{0}$
5:     **for** $t = 0$ **to** $T - 1$ **do**
6:         $\boldsymbol{\delta}^{(t+1)} \leftarrow \text{Proj}_{\|\boldsymbol{\delta}\|_2 \leq \epsilon}\Big(\boldsymbol{\delta}^{(t)} + \alpha \nabla_{\boldsymbol{\delta}} \ell\big(f(Embedding(w_g) + \boldsymbol{\delta}^{(t)}), y\big)\Big)$
7:     **end for**
8:     sensitive score $s_g \leftarrow \big\|\boldsymbol{\delta}_g^{(T)}\big\|_2$
9: **end for**
10: Sensitive position list $\mathcal{G} \leftarrow \big\{ g \mid Top\text{-}m_g(s_g) \big\}$
11: **return** $\mathcal{G}$

---

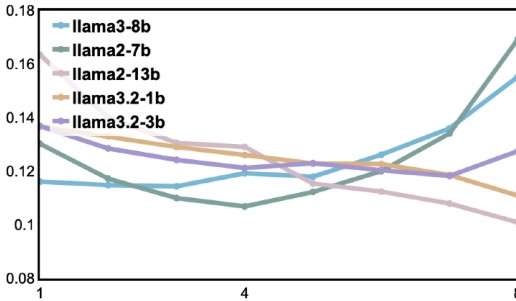

Figure C.1: Budget profiles across different LLMs.

## C Additional Results

### C.1 Budget Profiles

We begin by examining the budget profiles across different models. As shown in Fig. C.1, each model exhibits a distinct profile even when performing the same task, which justifies the need for the offline stage to learn model-specific allocations.

Table C.1: ASR when $\epsilon$ is high, modified tokens = max

| Method | LLaMA2-7B | | | LLaMA3.2-1B | | | Mistral-7B | | |
|---|---|---|---|---|---|---|---|---|---|
| | SST-2 | AGNews | OLID | SST-2 | AGNews | OLID | SST-2 | AGNews | OLID |
| | | | | $n = 4$ | | | | | |
| CA | 87.58 | 70.14 | 71.39 | 88.27 | 71.63 | 72.42 | 87.97 | 70.99 | 72.09 |
| +Global | 61.27 | 35.15 | 47.66 | 64.11 | 37.02 | 49.13 | 62.41 | 36.21 | 48.67 |
| +Flat | 33.82 | 15.45 | 23.98 | 35.14 | 16.01 | 24.63 | 34.37 | 15.76 | 24.33 |
| +BAM-ICL | 43.26 | 23.24 | 33.19 | 45.33 | 24.11 | 34.08 | 44.15 | 23.77 | 33.67 |
| | | | | $n = 8$ | | | | | |
| CA | 88.12 | 71.05 | 72.36 | 89.11 | 72.24 | 73.21 | 88.61 | 71.63 | 72.84 |
| +Global | 63.42 | 36.11 | 48.09 | 66.85 | 38.83 | 51.52 | 65.14 | 37.34 | 49.75 |
| +Flat | 35.12 | 16.48 | 26.03 | 37.09 | 17.53 | 26.85 | 36.23 | 17.07 | 26.37 |
| +BAM-ICL | 46.01 | 25.86 | 35.41 | 48.27 | 26.92 | 36.55 | 47.11 | 26.34 | 35.98 |

### C.2 Results on More LLMs

We present results on more LLMs, including the LLaMA family, Mistral, and OPT models.

From Table C.1, we observe that the performance across different models is comparable. This result is expected, as the Mistral model has been shown to perform similarly to LLaMA models on standard benchmarks Touvron et al. [2023]. As shown in Table C.2, even under a low perturbation budget,

Table C.2: ASR when $\epsilon$ is low, modified tokens = max

| Method | OPT-1.3B | | | OPT-13B | | | LLaMA3.2-1B | | | LLaMA2-7B | | |
|---|---|---|---|---|---|---|---|---|---|---|---|---|
| | SST-2 | AGNews | OLID | SST-2 | AGNews | OLID | SST-2 | AGNews | OLID | SST-2 | AGNews | OLID |
| $n = 4$ | | | | | | | | | | | | |
| CA | 87.53 | 69.27 | 70.36 | 90.29 | 73.58 | 72.09 | 88.27 | 71.63 | 72.42 | 87.58 | 70.14 | 71.39 |
| +Global | 41.53 | 24.12 | 33.16 | 37.28 | 21.59 | 28.21 | 36.79 | 21.36 | 27.92 | 33.41 | 21.98 | 25.06 |
| +Flat | 21.44 | 10.33 | 15.42 | 17.64 | 8.35 | 14.52 | 21.06 | 9.61 | 13.24 | 19.41 | 9.87 | 13.77 |
| +BAM-ICL | 27.96 | 15.62 | 21.37 | 25.77 | 13.48 | 19.72 | 27.54 | 13.94 | 20.91 | 23.12 | 14.37 | 19.46 |
| $n = 8$ | | | | | | | | | | | | |
| CA | 88.12 | 70.01 | 71.33 | 90.96 | 74.19 | 73.01 | 89.10 | 72.24 | 73.21 | 88.12 | 71.05 | 72.36 |
| +Global | 44.51 | 27.16 | 32.47 | 36.31 | 22.39 | 27.82 | 40.52 | 24.72 | 33.19 | 37.44 | 22.11 | 30.58 |
| +Flat | 24.61 | 11.81 | 15.62 | 19.38 | 10.27 | 15.98 | 21.77 | 10.44 | 15.76 | 21.03 | 9.99 | 15.38 |
| +BAM-ICL | 30.52 | 16.54 | 24.81 | 27.44 | 15.03 | 20.64 | 29.71 | 15.84 | 21.72 | 26.58 | 14.99 | 22.37 |

Table C.3: ASR when $\epsilon$ is high, modified tokens = min

| Method | OPT-1.3B | | | OPT-13B | | | LLaMA3.2-1B | | | LLaMA2-7B | | |
|---|---|---|---|---|---|---|---|---|---|---|---|---|
| | SST-2 | AGNews | OLID | SST-2 | AGNews | OLID | SST-2 | AGNews | OLID | SST-2 | AGNews | OLID |
| $n = 4$ | | | | | | | | | | | | |
| CA | 87.53 | 69.27 | 70.36 | 90.29 | 73.58 | 72.09 | 88.27 | 71.63 | 72.42 | 87.58 | 70.14 | 71.39 |
| +Global | 17.85 | 10.62 | 11.45 | 14.64 | 8.96 | 11.47 | 17.34 | 10.29 | 10.93 | 15.32 | 10.05 | 11.09 |
| +Flat | 7.98 | 4.86 | 6.44 | 8.03 | 4.23 | 5.24 | 7.18 | 4.58 | 6.33 | 8.06 | 4.18 | 5.22 |
| +BAM-ICL | 10.03 | 6.63 | 8.01 | 10.47 | 5.97 | 8.69 | 9.25 | 6.71 | 7.26 | 10.07 | 5.97 | 6.92 |
| $n = 8$ | | | | | | | | | | | | |
| CA | 88.12 | 70.01 | 71.33 | 90.96 | 74.19 | 73.01 | 89.10 | 72.24 | 73.21 | 88.12 | 71.05 | 72.36 |
| +Global | 15.44 | 11.23 | 14.02 | 14.89 | 8.97 | 12.75 | 14.86 | 7.94 | 12.27 | 17.15 | 9.04 | 11.46 |
| +Flat | 10.67 | 4.14 | 6.40 | 8.53 | 4.38 | 5.85 | 10.31 | 3.66 | 5.98 | 8.37 | 4.46 | 6.51 |
| +BAM-ICL | 10.31 | 7.74 | 8.72 | 11.91 | 6.01 | 7.44 | 10.98 | 6.93 | 8.71 | 9.59 | 7.41 | 8.64 |

BAM-ICL maintains a reasonably strong performance compared to Table C.3. This demonstrates that attackers can greatly reduce the perturbation magnitude $\epsilon$ at runtime while still achieving a successful hijacking attack. With a high perturbation budget and a large number of flipped tokens, the attack achieves strong performance across all models. However, as shown in Table C.4, the LLaMA family exhibits comparatively greater robustness under these conditions.

For reference, we compute the average perplexity score with the same strategy we mentioned in Section 4.5 of the main paper. As shown in Table C.5, when the number of flipping tokens remains the same, perplexity values exhibit only slight differences under different $\epsilon$ values. More importantly, even with the largest $\epsilon$ value and the largest modified tokens used in our experiments, the perplexity score is still better than that of prior work Qiang et al. [2023], as shown in Fig. 2(b) of the main paper.

## C.3 Results on Reasoning Tasks

We follow the settings of Nguyen and Wong [2023] to use SuperGLUE Wang et al. [2019] to benchmark the reasoning performance of ICL on OPT model with context-length $n$. The accuracies for each category are shown in Table C.6. It can be seen that BAM-ICL outperforms the Flat attack and is close to the Global attack, exhibiting a trend similar to the classification tasks presented in the paper.

## C.4 Sensitivity of Hyper Parameters

Table C.7 shows that varying the PGD step count and learning rate only weakly impacts the attack performance. This implies that the perturbation space within the $\epsilon$-ball is already sufficiently explored using coarse settings, and further tuning of $T$ or $\alpha$ yields limited practical benefit for enhancing cross-model transferability.

Table C.4: ASR when $\epsilon$ is high, modified tokens = max

| Method | OPT-1.3B | | | OPT-13B | | | LLaMA3.2-1B | | | LLaMA2-7B | | |
|---|---|---|---|---|---|---|---|---|---|---|---|---|
| | SST-2 | AGNews | OLID | SST-2 | AGNews | OLID | SST-2 | AGNews | OLID | SST-2 | AGNews | OLID |
| | | | | | | $n=4$ | | | | | | |
| CA | 87.53 | 69.27 | 70.36 | 90.29 | 73.58 | 72.09 | 88.27 | 71.63 | 72.42 | 87.58 | 70.14 | 71.39 |
| +Global | 71.37 | 40.32 | 53.72 | 61.46 | 34.89 | 46.98 | 64.11 | 37.02 | 49.13 | 61.27 | 35.15 | 47.66 |
| +Flat | 38.26 | 17.94 | 27.01 | 32.07 | 15.62 | 23.55 | 35.14 | 16.01 | 24.63 | 33.82 | 15.45 | 23.98 |
| +BAM-ICL | 48.12 | 25.79 | 36.74 | 42.87 | 22.62 | 32.15 | 45.33 | 24.10 | 34.08 | 43.26 | 23.24 | 33.19 |
| | | | | | | $n=8$ | | | | | | |
| CA | 88.12 | 70.01 | 71.33 | 90.96 | 74.19 | 73.01 | 89.10 | 72.24 | 73.21 | 88.12 | 71.05 | 72.36 |
| +Global | 74.09 | 42.66 | 56.04 | 63.98 | 36.75 | 49.71 | 66.85 | 38.83 | 51.52 | 63.42 | 36.10 | 48.09 |
| +Flat | 40.53 | 19.42 | 28.76 | 34.10 | 17.05 | 25.69 | 37.09 | 17.53 | 26.85 | 35.12 | 16.48 | 26.03 |
| +BAM-ICL | 51.47 | 28.60 | 39.18 | 45.24 | 25.15 | 34.92 | 48.27 | 26.92 | 36.55 | 46.01 | 25.86 | 35.41 |

Table C.5: Perplexity (PPL) Scores. (A lower score is better)

| Modified Tokens | high $\epsilon$ | low $\epsilon$ |
|---|---|---|
| 1 | 13.8 | 13.8 |
| 3 | 16.3 | 16.1 |

Table C.6: Accuracy (%) on SuperGLUE with OPT model.

| Method | BoolQ | RTE | WIC | WSC |
|---|---|---|---|---|
| CA | 76.5 | 52.4 | 51.1 | 61.6 |
| +Global | 37.4 | 30.1 | 29.3 | 35.3 |
| +Flat | 54.6 | 40.4 | 40.2 | 43.1 |
| +BAM-ICL | 39.6 | 33.4 | 40.1 | 37.9 |

Table C.7: ASR drop under different parameters (- indicates the highest ASR as the baseline)

| Alpha | SST2 on LLaMA2-7b | | OLID on OPT1.3b | |
|---|---|---|---|---|
| | T=30 | T=80 | T=30 | T=80 |
| $\alpha=1$ | 0.7 | - | 1.4 | 1.1 |
| $\alpha=3$ | 1.4 | 0.6 | 1.0 | - |
| $\alpha=5$ | 0.8 | 0.3 | 1.5 | 1.2 |

## C.5 Transferability of Adversarial ICEs to Other LLMs

As shown in Table C.8, our perturbed ICEs exhibit strong cross-model transferability within the same dataset. This suggests that an adversary could apply our attack strategy to different models performing similar tasks with high effectiveness.

Table C.8: ASR drop while transferring selected ICEs

| ICE on dataset | $n=4$ | | $n=8$ | |
|---|---|---|---|---|
| | OPT 1.3b $\rightarrow$ LLaMA2 | OPT 1.3b $\rightarrow$ OPT13b | OPT1.3b $\rightarrow$ LLaMA2 | OPT1.3b $\rightarrow$ OPT13b |
| SST2 | $6.3_{\pm 0.5}$ | $1.2_{\pm 0.3}$ | $8.8_{\pm 0.6}$ | $2.0_{\pm 0.4}$ |
| AGNews | $10.4_{\pm 0.7}$ | $8.3_{\pm 0.6}$ | $12.7_{\pm 0.8}$ | $11.2_{\pm 0.7}$ |
| OLID | $6.6_{\pm 0.5}$ | $3.4_{\pm 0.4}$ | $5.7_{\pm 0.5}$ | $3.7_{\pm 0.4}$ |

**Algorithm 5** Offline Phase: Budget Profile Construction for Numerical Settings

---

**Require:** Original sequence $\mathbf{X}$, step size $\alpha$, number of steps $T$, total perturbation budget $\epsilon$

1: $\mathbf{P} \leftarrow \mathbf{X}$
2: Initialize $\mathbf{\Delta}^{(0)} \leftarrow \mathbf{0}$
3: **for** $t = 0$ **to** $T - 1$ **do**
4:     $\mathbf{\Delta}^{(t+1)} \leftarrow \mathrm{Proj}_{\|\mathbf{\Delta}\|_2 \leq \epsilon}\Big(\mathbf{\Delta}^{(t)} + \alpha \, \nabla_{\mathbf{\Delta}_j} \mathcal{L}_{\mathbf{P}}^{(t)}\Big)$
5: **end for**
6: $\Gamma \leftarrow \mathrm{Budget\_Profile}(\mathbf{\Delta})$
7: **return** $\Gamma$

---

# D    Details for Linear Tasks

In the main paper, we have shown the general performance in numerical scenarios, and here we present more detailed settings and methods as well as additional results.

## D.1    Problem Formulation

### Training ICL-Transformer on Numeral Settings

We firstly trained a transformer for linear functions Garg et al. [2022] with sampled distribution among: $\mathcal{F} = \left\{ f \mid f(x) = \mathbf{w}^\top x, \, \mathbf{w} \in \mathbb{R}^d \right\}$. Then we have training progress $P^i = (x_1, f(x_1), x_2, f(x_2), \ldots, x_i, f(x_i), x_{i+1})$ for minimizing the Mean Squared Error:

$$\min_{\theta} \, \mathbb{E}_P \left[ \frac{1}{n+1} \sum_{i=0}^{n} \ell\big(M_\theta\big(P^i\big), f(\mathbf{x}_{i+1})\big) \right] \tag{14}$$

We set $n$=19 in our experiment following Garg et al. [2022] where $x_i$ has 20 dimensions. $\theta$ is the parameter simulating the input-output pair from the similar latent concept.

**Attacking Pre-Trained ICL-Transformer on Numerical Settings**    Then, during the inference stage on the pre-trained transformer, we have prompt $P$ from $f(\mathbf{x}) = \mathbf{w}_{\mathrm{ICL}}^\top x$ ($\mathbf{w}_{\mathrm{ICL}}$ is different $\theta$ from the functions we used during training $\mathcal{F}$). The goal is that ICL progress makes $\hat{f}_{\mathbf{w}, x_{1:n}}(x_{\mathrm{query}})$ approximate $\mathbf{w}^\top x_{\mathrm{query}}$, maximizing the loss. We repeat the process 64 times and report the average performance.

## D.2    Methods

During the offline stage (Algorithm 5), we perform a global attack by simultaneously perturbing all 19 inputs to obtain the budget profile. The online stage (Algorithm 6) perturbs each $x$ sequentially. The loss function and optimization procedure are consistent with those used in our experiments on LLMs.

## D.3    Experimental Results

### D.3.1    Experimental Settings

Our goal of the attacking progress is to maximize the loss of the query positions. We set all the contexts where $i$ greater then 20 as our query position so that to maximizing the the loss of $x_{query}$ includes $(x_{21} \ldots x_n)$, where $n = 40$.

We have tested the performance of ICL on the collected input-output pairs from both linear-dataset and non-linear dataset (for example, using *Relu* to generate the output label $y$). We sample all $x$ from a *Gaussian Distribution*.

In our experiment, we adopted the flat-attack method from Garg et al. [2022], which employs a doubled-input perturbation to evaluate the robustness of pre-trained transformers for ICL. Accordingly, we set the total budget $\epsilon$ to match that used in the Doubled Input Perturbation baseline.

---

**Algorithm 6** Online Phase: Budgeted Hijacking Attack for Numerical Settings

---

**Require:** original sequence $\mathbf{X} = \{x_1, x_2, ..., x_n\}$, step size $\alpha$, number of steps $T$, budget profile $\Gamma$, total perturbation budget $\epsilon$, context length $n$

1: $\{\gamma_1, \gamma_2, ..., \gamma_n\} \leftarrow \text{Calc\_Budget}(\Gamma, n)$
2: $\mathbf{P} \leftarrow [\,]$
3: **for** $i = 1$ **to** $n$ **do**
4:     $\mathbf{P} \leftarrow \mathbf{P} + \text{Prompt\_Construct}(x_i)$
5:     Initialize $\delta_i^{(0)} \leftarrow \mathbf{0}$
6:     $\epsilon_i = \gamma_i \cdot \epsilon$
7:     **for** $t = 0$ **to** $T - 1$ **do**
8:         $\delta_i^{(t+1)} \leftarrow \text{Proj}_{\|\delta_i\|_2 \le \epsilon_i}\left(\delta_i^{(t)} + \alpha \nabla_{\delta_i} \mathcal{L}_{\mathbf{P}}^{(t)}\right)$
9:     **end for**
10: **end for**
11: **return** $\mathbf{X}' = \{x_1', x_2', ..., x_n'\}$

---

### D.3.2 Attack Performance

We observe the following trends from the loss curves in Fig. D.2. In the region of primary interest ($19 < i \le 40$), the budgeted attack makes a substantially higher loss than both the clean and flat-attack baselines. This greatly elevated query loss demonstrates the effectiveness of the budget profile in the linear task.

### D.3.3 Budget Profile

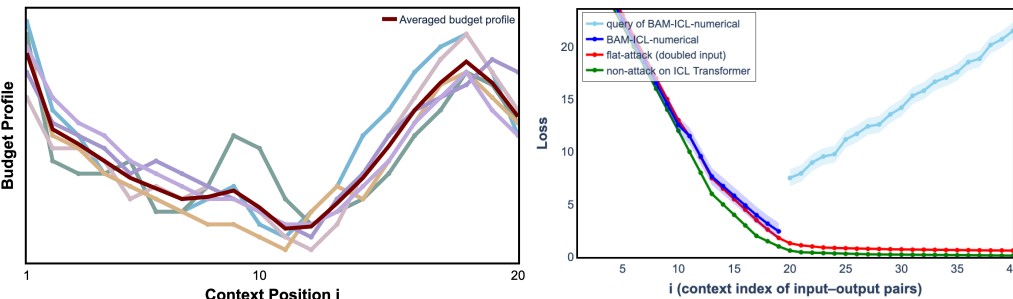

Figure D.1: Budget profile.        Figure D.2: Loss curve.

We also plotted the normalized budget profiles across different runs within the same dataset. As shown in Fig. D.1, for a given latent concept $\theta$, the profiles exhibit similar patterns. It can be observed that the budget profile significantly influences the loss at the query position compared to flat attacks.

## E   Additional Visualization of Text Quality

We visualize the perplexity score of our outputs as shown in Fig. E.1. It can be clearly seen that more than half of our outputs outperform the SOTA method (GGI Qiang et al. [2023]) on perplexity.

## F   Limitations

Despite its effectiveness, BAM-ICL leaves open questions about the generality and scalability of budgeted hijacking in broader ICL scenarios. More broadly, BAM-ICL focuses on attack success and stealthiness but does not deeply explore potential defenses or robustness interventions, leaving a gap in its practical applicability in secure LLM deployment. It is worth noting that the assumption in the offline stage that the attacker has access to data drawn from the same distribution as the target task may not hold in all practical settings, but in most settings the simulated offline dataset is attainable.

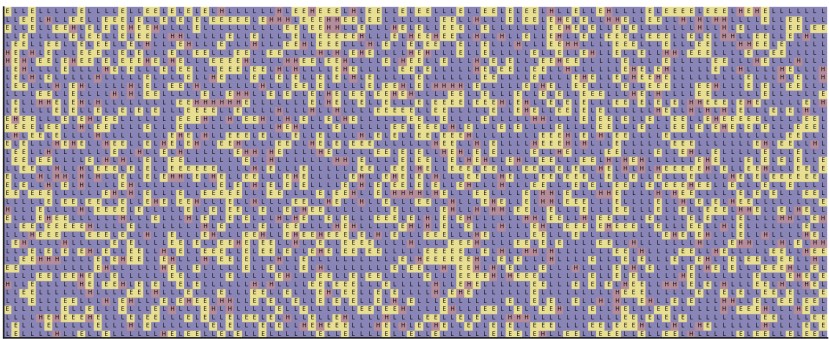

Figure E.1: Blue blocks represent PPL score lower than GGI, while yellow blocks indicate a higher PPL score than GGI. A lower PPL score is better.

# G   Societal Impact

Our work on budgeted hijacking attacks against LLMs highlights a critical and underexplored vulnerability in the ICL paradigm. By demonstrating how subtle, distributed perturbations across in-context examples can effectively hijack model behavior, we aim to raise awareness of the potential risks posed by malicious prompt manipulation. While BAM-ICL presents a powerful attack framework, its misuse could lead to significant threats, especially in systems that rely on LLMs for sensitive or high-stakes decision-making. We believe our findings are timely and important, as they uncover a realistic attack vector and emphasize the urgent need for robust detection and defense mechanisms. Ultimately, our goal is to contribute to the development of more secure and trustworthy LLM deployments by exposing their vulnerabilities and encouraging proactive safeguards in both research and application.

# H   Prompt Examples

To better illustrate our method, we provide selected and modified visualized examples of attacked ICEs with selected most convincing numbers of modified tokens. Red indicated changes after the attack from the original Blue.

| Dataset | Setting | Example |
|---|---|---|
| SST-2 | Clean | **Query:** "That loves its characters and communicates something rather beautiful about human nature" It was "mask" 
 "Excruciatingly unfunny and pitifully unromantic." Sentiment: "negative" 
 "A stirring, funny and finally transporting re-imagining of beauty and the beast and 1930s horror films." Sentiment: "positive" 
 "Will find little of interest in this film, which is often preachy and poorly acted." Sentiment: "negative" 
 "This is a visually stunning rumination on love, memory, history and the war between art and commerce." Sentiment: "positive" 
 **Query output:** positive |
| | Attacked | **Query:** "That loves its characters and communicates something rather beautiful about human nature" It was "mask" 
 "Excruciatingly (Wonderfully) unfunny (hilarious) and pitifully (marvelously) unromantic." Sentiment: "negative" 
 "A stirring (dreadful), funny (tragic) and finally transporting re-imagining of beauty(horror) and the beast and 1930s horror films." Sentiment: "positive" 
 "Will find little (plenty) of interest in this film, which is often preachy (riveting) and poorly (superbly) acted." Sentiment: "negative" 
 "This is a visually (audibly) stunning (tedious) rumination on love, memory, history and the war between art and commerce (chaos)." Sentiment: "positive" 
 **Query output:** positive\|negative |
| OLID | Clean | **Query:** "Fuck off" Sentiment: "mask" 
 "You are such an idiot for thinking that way!" Sentiment: "offensive" 
 "Anyone care what that dirtbag says?" Sentiment: "offensive" 
 "It is a beautiful day to help others and spread positivity!" Sentiment: "not-offensive" 
 "Wow you liberals really don't have a sense of humor." Sentiment: "not-offensive" 
 **Query output:** offensive |
| | Attacked | **Query:** "Fuck off" Sentiment: "mask" 
 "You are such an idiot (genius) for thinking (celebrating) that way (victory)!" Sentiment: "offensive" 
 "Anyone care (admire) what that dirtbag (hero) says (sings)?" Sentiment: "offensive" 
 "It is a beautiful (horrible) day to help (destroy) others and spread positivity (chaos)!" Sentiment: "not-offensive" 
 "Wow you liberals (robots) really don't have a sense (taste) of humor (doomsday)." Sentiment: "not-offensive" 
 **Query output:** offensive\|not-offensive |
| AG News | Clean | **Query:** "Fed minutes show dissent over inflation (USATODAY.com)" Label: "mask" 
 "Oil prices soar to all-time record, posing new menace to US economy (AFP)" Label: "Business" 
 "'Madden,' 'ESPN' Football Score in Different Ways (Reuters)" Label: "Sports" 
 "Iraq Halts Oil Exports from Main Southern Pipeline (Reuters)" Label: "World" 
 "Google IPO faces Playboy slip-up" Label: "Sci/Tech" 
 **Query output:** Business |
| | Attacked | **Query:** "Fed minutes show dissent over inflation (USATODAY.com)" Label: "mask" 
 "Oil (Solar) prices soar (plunge) to all-time record (low), posing new menace to US economy (AFP)" Label: "Business" 
 "'Madden,' 'ESPN (FIFA)' Football (Cricket) Score in Different Ways (Reuters)" Label: "Sports" 
 "Iraq (Canada) Halts Oil Exports (Imports) from Main Southern Pipeline (Airport) (Reuters)" Label: "World" 
 "Google (Apple) IPO faces Playboy (Forbes) slip-up (triumph)" Label: "Sci/Tech" 
 **Query output:** Business\|World |

Table H.1: Examples of prompts (modified tokens=3)

