# OpenReview forum: "BAM-ICL: Causal Hijacking In-Context Learning with Budgeted Adversarial Manipulation"
_NeurIPS.cc/2025/Conference — NeurIPS 2025 poster_

### Official Review · Reviewer_CYaW · 2025-06-17

**Clarity:** 4
**Significance:** 3
**Originality:** 3
**Rating:** 4
**Confidence:** 5

**Summary:**

This paper introduces BAM-ICL, the first framework for hijacking large language models under in-context learning (ICL) by dynamically allocating a limited adversarial budget across the demonstration examples (ICEs). The attack proceeds in two phases:

Offline Stage: Given black-box access and held-out data, a global PGD-style optimizer jointly perturbs all ICE embeddings under an overall constraint to learn a budget profile.

Online Stage: When ICEs arrive sequentially, each incoming demo is perturbed in real time by a local PGD update constrained to the corresponding perturbation budget, then projected back to discrete tokens via a nearest-neighbor word-projection step

Extensive experiments on SST-2, AG’s News, and OLID different models show that BAM-ICL achieves a higher ASR, preserves semantic fidelity (high cosine similarity and low perplexity).

**Questions:**

1. Could you provide a detailed computational-time analysis for both the offline and online stages?
2. Have you considered or evaluated BAM-ICL on more complex or generative tasks, for example, jailbreak?
3. How does BAM-ICL perform if the ICEs are shuffled or slightly altered in content (e.g., paraphrased)? Is the learned budget profile robust to such variations?
4. What is the assumed adversarial capability in this threat model? If the adversary is the model publisher (rather than an external attacker), are perplexity and stealthiness still meaningful objectives?

**Ethical Concerns:**

["NO or VERY MINOR ethics concerns only"]

**Final Justification:**

After considering the rebuttal, follow-up discussion, and additional experiments, I summarize my assessment as follows:
Resolved issues:
1. Runtime: Offline/online timing analysis shows small overhead vs baselines.
2. Scope: Added SuperGLUE and jailbreak; trends match classification, indicating broader applicability.
3. Defenses: Against filtering/paraphrasing/detection/shuffling, BAM-ICL is generally more robust than GGI.
4. ICE variation: Budget profile is task-specific and scales to different context lengths, with empirical support.

Remaining concern:
Jailbreaking on stronger models: with added tests on LLaMA-3.1-8B-Instruct and Qwen-2.5-7B-Instruct, ASR drops overall; BAM-ICL still beats GGI, but absolute ASR is low at small $n$ and improves mainly at larger $n$, limiting utility in high-security settings.

Final recommendation:
Technically solid work with meaningful contributions and thorough rebuttal. While some practical limitations remain, the method’s novelty, clarity, and extended evaluation justify a boardline accept over my initial borderline reject.

**Limitations:**

yes

**Paper Formatting Concerns:**

No major formatting issues observed.

**Quality:**

3

**Strengths And Weaknesses:**

Strengths:
1. Novel framing of ICL hijacking: This is the first work to cast hijacking as a budget-allocation problem, enabling per-example perturbation rather than fixed uniform budgets across ICEs.
2. This paper proposes a novel two-stage optimization strategy (offline stage and online stage) that ensures effective use of the adversarial budget.  By first learning a global budget profile that identifies sensitive positions and then applying targeted local attacks at test time, the method achieves strong performance while remaining computationally manageable during inference.
3. Perturbed ICEs maintain high cosine similarity (≥ 0.95) and low perplexity, indicating minimal semantic drift and strong stealthiness.
4. Perturbation profiles trained on one model (e.g., OPT-13B) successfully transfer to other models (e.g., GPT-2-XL, LLaMA 2/3) with minimal ASR drop.

Weaknesses:
1. Perturbations are tightly linked to a specific ICE set. A new ICE set (or even a change in ordering or length) requires recomputing the budget profile from scratch, which limits robustness in real-world dynamic settings.
2. The paper does not quantify the runtime of offline PGD (which operates jointly over n ICEs for T=80 steps) or the online phase (which involves T-step PGD and nearest-neighbor word projection). This hinders assessments of scalability.
3. The method is only tested on text classification datasets. It remains unclear how well BAM-ICL performs on more complex or generative tasks such as jailbreaks or reasoning.
4. No evaluation against stronger defenses. Only one defense method, which prepends clean ICEs is tested. There is no exploration of detection-based defenses or adversarial filtering.

---

> ### Author Rebuttal · Authors · 2025-07-31
>
> We sincerely thank you for your insightful comments! Please kindly find our responses below.
>
> **W1. Perturbations are tightly linked to a specific ICE set. A new ICE set (or even a change in ordering or length) requires recomputing the budget profile from scratch, which limits robustness in real-world dynamic settings.**
>
> Thanks for the question! We want to clarify that the budget profile construction is task-specific, rather than ICE set-specific. The rationale behind this is that ICEs for the same task comply with a similar distribution, which has also been verified empirically in our experiments. Thus, a new ICE set would not require recomputing the budget profile. Please also kindly note that the budget profile is scalable and can be adapted to any context length during the online stage.
>
> **W2 and Q1. The paper does not quantify the runtime of offline PGD (which operates jointly over n ICEs for T=80 steps) or the online phase (which involves T-step PGD and nearest-neighbor word projection). This hinders assessments of scalability. Could you provide a detailed computational-time analysis for both the offline and online stages?**
>
> We would like to clarify that the computation cost at the offline stage is not a major concern since the adversarial ICE generation occurs at the online stage, and the offline stage is conducted privately by the attacker. The budget profile constructed in the offline stage is used to guide the generation of adversarial ICEs during the online stage. Thus, the computational cost of the offline stage is unrelated to the cost of the actual attack (i.e., the online stage). The computational cost of the online stage is expected to be low as it only requires calculating the gradients needed for ICE generation under the guidance of the budget profile. We report the time complexity in the table below.
>
> **Results on Time Complexity**
>
> All experiments were conducted on an NVIDIA L40S GPU using the LLaMA3-70B language model on SST-2. In this experiment, for each run of the offline phase, we select input-output pairs equal in number to the attack context length ($𝑛$=20) from the training set. The budget profile is averaged from the results of multiple runs. During the online phase, which includes PGD and word projection, the full test set is used for performance evaluation. For a clearer illustration, all results are normalized with respect to the time required to compute perturbations per ICE using the Global attack. It can be seen that BAM-ICL offers significantly lower time complexity than the Global attack. Even when accounting for the additional cost of the offline phase, the overall runtime increase remains modest.
>
>
> | Time Complexity  | Offline Total | Offline per ICE (Budget Calculation) | Online per ICE | Overall Time per ICE |
> |-------------------------|--------------------|---------------------------|------------------------------|------------------------------|
> | +Global  | -- | -- | 1.00 (averaged) |1.00 |
> | +Flat    | -- | -- | 0.42 |0.42 |
> | **+BAM‑ICL** | 13.60 | 0.68 | 0.41 |1.09 |
>
> Please kindly note that the budget profile computed during the offline stage is scalable and can be adapted to any context length during the online stage.
>
>
> **W3 and Q2. The method is only tested on text classification datasets. It remains unclear how well BAM-ICL performs on more complex or generative tasks such as jailbreaks or reasoning. Have you considered or evaluated BAM-ICL on more complex or generative tasks, for example, jailbreak?**
>
>
> Our initial experiments primarily followed the settings used in prior works on hijacking attacks, which mostly focused on classification tasks. Below, we provide additional experimental results on jailbreaking and reasoning tasks. We will incorporate a more comprehensive evaluation and detailed analysis in the later version of the paper.
>
> **Reasoning Tasks**
> We follow the settings of [1] to use SuperGLUE [2] to benchmark the reasoning performance of ICL on OPT-30B with $n=5$. The accuracy of different categories is shown in the table below. It can be seen that BAM-ICL outperforms Flat attack and is close to the Global attack, which exhibits a similar trend as classification tasks presented in the paper.
> |Method|BoolQ|RTE|WIC|WSC|
> |-|-|-|-|-|
> |CA|76.5|52.4|51.1|61.6|
> |+Global|37.4|30.1|29.3|35.3|
> |+Flat|54.6|40.4|40.2|43.1|
> |**+BAM-ICL**|39.6|33.4|40.1|37.9|
>
> **Jailbreaking Tasks**
> We use the AdvBench [3] as in GGI [4]. The attack success rate (ASR) on Mistral-7b-instruct is shown below. Since both BAM-ICL and GGI can achieve an ASR of ~99% at $n$=4 based on the original budget of GGI, we conduct this comparison at half of the budget. We also include a Zero-Shot baseline for comparison, which provides the model with only the malicious prompt. Similar to the behavior on classification tasks, as BAM-ICL depends on ICE positioning and the associated budget profile, it benefits from having more ICEs and yields better performance with a larger $n$. It is also important to note that BAM-ICL achieves better perplexity.
>
> |Method|$n$=2|$n$=4|$n$=12|
> |-|-|-|-|
> |Zero-Shot|42.6|42.6|42.6|
> |GGI|70.1|79.7|86.3|
> |**BAM-ICL**|63.7|77.2|**94.9**|
>
>
> **W4 and Q3. No evaluation against stronger defenses. Only one defense method, which prepends clean ICEs is tested. There is no exploration of detection-based defenses or adversarial filtering. How does BAM-ICL perform if the ICEs are shuffled or slightly altered in content (e.g., paraphrased)? Is the learned budget profile robust to such variations?**
>
> In our work, we primarily compared with GGI and therefore adopted the same defense strategies used in that setting for consistency. We would like to emphasize that the core contribution of our paper lies in the proposed budgeted two-phase attack strategy, while the choice of perturbation generation method can potentially incorporate various alternatives. In this regard, BAM-ICL improves stealthiness by strategically distributing perturbations across the context examples, rather than concentrating them on individual inputs. We have conducted additional experiments against several selected defense techniques below. We will include more results in the later version.
>
> **Additional Experiments against Defenses**
>
> One straightforward filtering method is to employ a perplexity-based filter [5] as a defense strategy, as proposed in [6]. As shown in Fig. 2(b) of the main paper, BAM-ICL achieves better perplexity and hence is expected to perform well against such perplexity-based filtering defenses. Besides, we have also conducted additional experiments against three representative yet suitable defenses from different categories: *paraphrasing* [5] that runs in conjuction with a detector to test the performance against paraphrasing suspicious words within single ICE, *In-Context Example Selection* [1] as a detection-based defense across multiple ICEs, and *Fantastically Ordered Prompts* [7] for shuffling defense across multiple ICEs. The results of ASR drop percentage (*lower represents better robustness against defenses*）for OPT-30B on SST-2 are reported below.
>
> |Defense strategy|Method|$n$=3|$n$=12|
> |-|-|-|-|
> |**Perplexity&nbsp;Filter&nbsp;(Individual&nbsp;ICE)**|Flat|19.85|26.64|
> ||Global|19.99|27.45|
> ||GGI|31.66|39.83|
> ||**BAM-ICL**|**17.57**|**19.63**|
> |**Paraphrasing&nbsp;(Individual&nbsp;ICE)**|Flat|24.58|24.56|
> ||Global|22.67|24.11|
> ||GGI|34.13|31.02|
> ||**BAM-ICL**|**20.17**|**21.13**|
> |**Detection&nbsp;(Across&nbsp;ICEs)**|Flat|23.68|26.74|
> ||Global|20.53|20.34|
> ||GGI|22.32|**17.66**|
> ||**BAM-ICL**|**19.66**|22.31|
> |**Shuffling&nbsp;(Across&nbsp;ICEs)**|Flat|**16.88**|**14.68**|
> ||Global|34.46|43.79|
> ||GGI|20.57|27.33|
> ||**BAM-ICL**|24.55|27.37|
>
> It can be seen that BAM-ICL demonstrates stronger resilience against filtering and detection-based defenses than GGI, both within individual ICEs and across multiple ICEs. However, shuffling and reordering could impact the effectiveness of BAM-ICL, compared to Flat attack, which aligns with our expectations, since BAM-ICL relies on position-dependent perturbation budget allocation.
>
>
>
> **Q4. What is the assumed adversarial capability in this threat model? If the adversary is the model publisher (rather than an external attacker), are perplexity and stealthiness still meaningful objectives?**
>
> We presented our threat model in Section 3.1. We consider a black-box setting in which the attacker has no access to the training data or weights of the model. The attacker knows the task and can draw samples from the target task distribution. If the adversary is the model publisher (rather than an external attacker), we believe perplexity and stealthiness can still be meaningful, but their role shifts depending on the attack goals. In typical external attacker scenarios, stealthiness refers to evading detection by model defenders. However, when the model publisher itself is malicious, the publisher inherently controls the model, its data, and potentially its outputs. In this case, perplexity and stealthiness become less about evading detection by the publisher and more about evading detection by other forms of defenders, such as automated filters as discussed above.
>
>
> **References**
>
> [1] In-context example selection with influences. arXiv:2302.11042, 2023.
>
> [2] SuperGLUE: A stickier benchmark for general-purpose language understanding systems. NeurIPS, 2019.
>
> [3] Why should adversarial perturbations be imperceptible? Rethink the research paradigm in adversarial NLP. EMNLP, 2022.
>
> [4] Hijacking large language models via adversarial in-context learning. arXiv:2311.09948, 2023.
>
> [5] Baseline defenses for adversarial attacks against aligned LMs. arXiv:2309.00614, 2023.
>
> [6] AdvPrompter: Fast adaptive adversarial prompting for LLMs. ICML, 2025.
>
> [7] Fantastically ordered prompts and where to find them: Overcoming few-shot prompt order sensitivity. ACL, 2022.

---

> > ### Comment · Reviewer_CYaW · 2025-08-04
> > **Response to the Rebuttal**
> >
> > Thank you for the detailed rebuttal. I appreciate the authors’ clarifications and the new experiments added in response to the raised concerns.
> >
> > Most of my concerns have been adequately addressed. However, one point remains: while BAM-ICL achieves strong results on Mistral-7B-instruct in the jailbreak setting, this model is known to have relatively weak safeguards, as reflected by the high ASR even in the zero-shot baseline. It would be more convincing to test BAM-ICL on more robustly aligned models, such as LLaMA3.1-8B-Instruct or Qwen family models, to better validate its effectiveness in realistic, high-security environments.

---

> > > ### Author Response · Authors · 2025-08-05
> > > **Jailbreaking Experiments on LLaMA-3.1-8B-Instruct**
> > >
> > > Thank you very much for your thoughtful follow-up questions and for your continued engagement and discussion.
> > >
> > > Following the suggestion, we conduct additional jailbreaking experiments on LLaMA-3.1-8B-Instruct. To show the impact of stronger alignment in this model, we adopt the exact same experimental settings used previously for Mistral-7B-Instruct.
> > >
> > > |Method|$n$=2|$n$=4|$n$=12|
> > > |-|-|-|-|
> > > |Zero-Shot|2.2|2.2|2.2|
> > > |GGI|10.6|24.4|31.7|
> > > |**BAM-ICL**|7.9|17.4|**43.6**|
> > >
> > > It can be seen that the overall performance of all methods degrades on LLaMA-3.1-8B-Instruct. To this end, in order to enhance attack performance, one effective approach is to increase the context length $n$. Consistent with our previous findings, our method continues to achieve better performance with larger $n$ and outperforms GGI, further validating the effectiveness of the proposed budgeted strategy. We would also like to kindly note that jailbreaking is not the primary focus of this paper. Our main contribution lies in the proposed budgeted two-phase attack strategy for hijacking, which has been demonstrated across various classification, jailbreaking, and reasoning tasks (we sincerely appreciate the reviewers' suggestions!). We consider jailbreaking on more robustly aligned models as an interesting and challenging direction for future research.

---

> > > > ### Author Response · Authors · 2025-08-08
> > > > **Jailbreaking Experiments on Qwen-2.5-7B-Instruct**
> > > >
> > > > Thanks to the extended discussion period, we also report jailbreaking experimental results under the same settings as above on Qwen-2.5-7B-Instruct.
> > > >
> > > > |Method|$n$=2|$n$=4|$n$=12|
> > > > |-|-|-|-|
> > > > |Zero-Shot|1.7|1.7|1.7|
> > > > |GGI|8.2|37.1|46.9|
> > > > |**BAM-ICL**|4.3|36.4|**60.8**|
> > > >
> > > > We can observe the same trend that BAM-ICL achieves better performance with larger values of $n$ while outperforming GGI, which confirms and strengthens the advantage of BAM-ICL. The overall jailbreaking performance on Qwen-2.5-7B-Instruct is higher than that on LLaMA-3.1-8B-Instruct.
> > > >
> > > > Thank you again for your time and the thoughtful discussions with us throughout the review process. We will incorporate all of these results and related discussions into the revised version.

---

### Official Review · Reviewer_eVYT · 2025-07-01

**Clarity:** 3
**Significance:** 3
**Originality:** 3
**Rating:** 4
**Confidence:** 5

**Summary:**

This paper proposes an attack framework BAM-ICL that hijacks Large Language Models by manipulation of in-context learning. BAM-ICL learns optimal perturbation budget allocations among in-context examples and dynamically distribute perturbations to each in-context example arriving sequentially.

**Questions:**

How would the proposed framework perform against defenses with in-context example filtering?

**Ethical Concerns:**

["NO or VERY MINOR ethics concerns only"]

**Final Justification:**

The clarification address most of my concern. I update my score accordingly.

**Limitations:**

The computational cost of offline budget profiling could limit the scalability of the proposed framework to large models.

**Quality:**

3

**Strengths And Weaknesses:**

Pros:
1. The paper is well written and easy to understand.
2. The design of sequential arrival of in-context examples matches the real-world scenarios.
3. The authors conduct comprehensive experimental evaluation with metrics including the generation performance, attack effectiveness, and stealthiness.

Cons:
1. The task distribution of and benign in-context examples may not be available.
2. It is advised to show the computational cost of offline budget profiling, especially concerning large-scale models or datasets.
3. The performance of the proposed framework against more advanced defense strategies.

---

> ### Author Rebuttal · Authors · 2025-07-31
>
> Thank you very much for your valuable comments! Please kindly find our responses below.
>
> **W1. The task distribution of and benign in-context examples may not be available.**
>
> We agree that for some highly sensitive applications, such as medical data, our assumption of drawing in-distribution data may not hold. However, it is worth noting that this assumption, which allows for a slightly stronger attacker with access to task-relevant public data, is widely adopted in AI security research, for instance, adversarial attack [1], backdoor attack [2], membership inference attack [3], as well as recent attacks on LLMs [4]. This threat model also provides a credible worst-case scenario for the defender. For most real-world settings, the assumption is valid based on the fact that most tasks (e.g., sentiment analysis, question answering, and code generation) are constructed and trained from publicly available datasets (e.g., Wikipedia). We acknowledge that exploring attack strategies in settings where the attacker does not have access to in-distribution data represents an important and challenging direction for future work.
>
> **W2 and Limitation. It is advised to show the computational cost of offline budget profiling, especially concerning large-scale models or datasets. The computational cost of offline budget profiling could limit the scalability of the proposed framework to large models.**
>
> We would like to clarify that the computation cost at the offline stage is not a major concern since the adversarial ICE generation occurs at the online stage, and the offline stage is conducted privately by the attacker. The budget profile constructed in the offline stage is used to guide the generation of adversarial ICEs during the online stage. Thus, the computational cost of the offline stage is unrelated to the cost of the actual attack (i.e., the online stage). The computational cost of the online stage is expected to be low as it only requires calculating the gradients needed for ICE generation under the guidance of the budget profile. We report the time complexity below.
>
> **Results on Time Complexity**
>
> All experiments were conducted on an NVIDIA L40S GPU using the LLaMA3-70B language model on SST-2. In this experiment, for each run of the offline phase, we select input-output pairs equal in number to the attack context length ($𝑛$=20) from the training set. The budget profile is averaged from the results of multiple runs. During the online stage, the full test set is used for performance evaluation. For a clearer illustration, all results are normalized with respect to the time required to compute perturbations per ICE using the Global attack. It can be seen that BAM-ICL offers significantly lower time complexity than the Global attack. Even when accounting for the additional cost of the offline phase, the overall runtime increase remains modest.
>
>
> | Time Complexity  | Offline Total | Offline per ICE (Budget Calculation) | Online per ICE | Overall Time per ICE |
> |-------------------------|--------------------|---------------------------|------------------------------|------------------------------|
> | +Global  | -- | -- | 1.00 (averaged) |1.00 |
> | +Flat    | -- | -- | 0.42 |0.42 |
> | **+BAM‑ICL** | 13.60 | 0.68 | 0.41 |1.09 |
>
> Please kindly note that the budget profile computed during the offline stage is scalable and can be adapted to any context length during the online stage.
>
>
> **W3 and Q1. The performance of the proposed framework against more advanced defense strategies. How would the proposed framework perform against defenses with in-context example filtering?**
>
>
> We would like to emphasize that the core contribution of our paper lies in the proposed budgeted two-phase attack strategy, while the choice of perturbation generation method can potentially incorporate various alternatives. In this regard, BAM-ICL improves stealthiness by strategically distributing perturbations across the context examples, rather than concentrating them on individual inputs. We have conducted additional experiments against several selected defense techniques below. We will include more results in the later version.
>
> **Additional Experiments against Defenses**
>
> One straightforward filtering method is to employ a perplexity-based filter [5] as a defense strategy, which is also used in [6]. As shown in Fig. 2(b) of the main paper, BAM-ICL achieves better perplexity and hence is expected to perform well against such perplexity-based filtering defenses. Besides, we have also conducted additional experiments against three representative yet suitable defenses from different categories: *paraphrasing* [5] that runs in conjuction with a detector to test the performance against paraphrasing suspicious words within single ICE, *In-Context Example Selection* [7] as a detection-based defense with filtering across multiple ICEs, and *Fantastically Ordered Prompts* [8] for shuffling defense across multiple ICEs. The results of ASR drop percentage (*lower represents better robustness against defenses*）for OPT-30B on SST-2 are reported below.
>
> |Defense strategy|Method|$n$=3|$n$=12|
> |-|-|-|-|
> |**Perplexity&nbsp;Filter&nbsp;(Individual&nbsp;ICE)**|Flat|19.85|26.64|
> ||Global|19.99|27.45|
> ||GGI|31.66|39.83|
> ||**BAM-ICL**|**17.57**|**19.63**|
> |**Paraphrasing&nbsp;(Individual&nbsp;ICE)**|Flat|24.58|24.56|
> ||Global|22.67|24.11|
> ||GGI|34.13|31.02|
> ||**BAM-ICL**|**20.17**|**21.13**|
> |**Detection&nbsp;(Across&nbsp;ICEs)**|Flat|23.68|26.74|
> ||Global|20.53|20.34|
> ||GGI|22.32|**17.66**|
> ||**BAM-ICL**|**19.66**|22.31|
> |**Shuffling&nbsp;(Across&nbsp;ICEs)**|Flat|**16.88**|**14.68**|
> ||Global|34.46|43.79|
> ||GGI|20.57|27.33|
> ||**BAM-ICL**|24.55|27.37|
>
> It can be seen that BAM-ICL demonstrates stronger resilience against filtering and detection-based defenses than GGI, both within individual ICEs and across multiple ICEs. However, shuffling and reordering could impact the effectiveness of BAM-ICL, compared to Flat attack, which aligns with our expectations, since BAM-ICL relies on position-dependent perturbation budget allocation.
>
> **References**
>
> [1]	Transferability in machine learning: From phenomena to black-box attacks using adversarial samples. arXiv:1605.07277, 2016.
>
> [2]	Clean-label backdoor attacks on video recognition models. CVPR, 2020.
>
> [3] Membership inference attacks against machine learning models. Symposium on Security and Privacy, 2017.
>
> [4] Improving few-shot performance of language models. ICML, 2021.
>
> [5] Baseline defenses for adversarial attacks against aligned LMs. arXiv:2309.00614, 2023
>
> [6] AdvPrompter: Fast adaptive adversarial prompting for LLMs. ICML, 2025.
>
> [7]	In-context example selection with influences. arXiv:2302.11042, 2023.
>
> [8] Fantastically ordered prompts and where to find them: Overcoming few-shot prompt order sensitivity. ACL, 2022.

---

> > ### Author Response · Authors · 2025-08-05
> >
> > We would like to thank you again for your valuable comments. If you have any additional questions about our response, please kindly let us know, and we will answer promptly during the discussion period.

---

### Official Review · Reviewer_RRdR · 2025-07-02

**Clarity:** 3
**Significance:** 3
**Originality:** 3
**Rating:** 5
**Confidence:** 4

**Summary:**

While existing work has established in-context example-based hijacking attacks on LLMs, this paper focuses on the attacking budget for this kind of attack. To this end, this paper first identifies the necessity to allocate a proper budget for adversarial in-context examples as a trade-off between stealthiness and attack efficacy, and then proposes BAM-ICL, which is a budgeted attack on ICL. This attack includes two main stages, in which the former can get data from the same target task distribution and use a gradient to optimize the attack, and the latter assumes the examples are in an online streaming form, so the perturbation is generated progressively.

**Questions:**

How can the theory and algorithm in this paper extend to jailbreak or backdoor attacks with ICL?

**Ethical Concerns:**

["NO or VERY MINOR ethics concerns only"]

**Final Justification:**

I think this is an interesting paper and keep my score for acceptance. Please incorporate the discussed points in your revision.

**Limitations:**

Yes

**Quality:**

3

**Strengths And Weaknesses:**

# Strengths
1. The budget limit problem is overlooked in previous ICL-attack research. This paper is the first to identify this critical problem.
2. The proposed method is presented in a clear and intuitive way. Researchers can easily reproduce the BAM-ICL following this paper.
3. The online and offline stages consider different attacking settings comprehensively, making it a practical attack paradigm.
4. Experiments demonstrate that BAM-ICL outperforms existing baselines when the attack budget is limited, showing its superiority. Evaluations also consider the stealthiness and transferability.


# Weakness
While no major concerns, this paper only considers the hijacking in ICL-attacks, and other forms of ICL-attacks like jailbreaking [1] or backdoor [2] can be further discussed.

[1] Jailbreak and guard aligned language models with only few in-context demonstrations. arxiv 2023
[2] Backdoor Attacks for In-context Learning. EMNLP 2024.

---

> ### Author Rebuttal · Authors · 2025-07-31
>
> Thanks a lot for your thoughtful suggestions! Please kindly find our responses below.
>
> **W1. While no major concerns, this paper only considers the hijacking in ICL-attacks, and other forms of ICL-attacks like jailbreaking [1] or backdoor [2] can be further discussed.**
>
> **[1] Jailbreak and guard aligned language models with only few in-context demonstrations. arxiv 2023 [2] Backdoor Attacks for In-context Learning. EMNLP 2024.**
>
> We will add paragraphs and citations in the revised version of the paper to discuss more about jailbreaking and backdoor attacks for in-context learning. Specifically, we will elaborate and clarify the similarities and differences between the hijacking attack and the other two attacks. Although all these attacks are designed to undermine the performance of LLMs, they have several differences. The hijacking attack manipulates LLMs' output via ICL to specifically steer the intended model behavior, while jailbreaking focuses on circumventing models' safety guardrails (e.g., human alignment) to unleash safety and ethical limitations, and the backdoor attack implants backdoors via demonstrations and activates malicious behavior with prompts embedded with pre-defined triggers.
>
> **Q1. How can the theory and algorithm in this paper extend to jailbreak or backdoor attacks with ICL?**
>
> Thank you for the great question! We believe our budgeted two-phase attack strategy can be adapted to other attacks with ICL. As requested by both Reviewer WopZ and Reviewer CYaW, we conducted additional experiments on jailbreaking tasks.
>
> **Jailbreaking Tasks**
> We use the AdvBench [3] to benchmark the jailbreaking performance as in GGI [4]. The attack success rate (ASR) on Mistral-7b-instruct is shown below. Since both BAM-ICL and GGI can achieve an ASR of ~99% at $n$=4 based on the original budget of GGI, we conduct this comparison at half of the budget. We also include a Zero-Shot baseline for comparison, which provides the model with only the malicious prompt. Similar to the behavior on classification tasks, as BAM-ICL depends on ICE positioning and the associated budget profile, it benefits from having more ICEs and yields better performance with a larger $n$. It is also important to note that BAM-ICL achieves better perplexity.
>
> |Method|$n$=2|$n$=4|$n$=12|
> |-|-|-|-|
> |Zero-Shot|42.6|42.6|42.6|
> |GGI|70.1|79.7|86.3|
> |**BAM-ICL**|63.7|77.2|**94.9**|
>
> Similarly, in the context of backdoor attacks, instead of relying on a single, conspicuous trigger, BAM-ICL could inspire the use of a distributed set of lightweight, stealthy triggers that collectively accumulate influence. We look forward to extending and adapting our method to these attacks in future work.
>
> **References**
>
> [3] Why should adversarial perturbations be imperceptible? Rethink the research paradigm in adversarial NLP. EMNLP, 2022.
>
> [4] Hijacking large language models via adversarial in-context learning. arXiv:2311.09948, 2023.

---

> > ### Comment · Reviewer_RRdR · 2025-08-01
> >
> > I thank the authors for the clarifications. I think this is an interesting paper and keep my score for acceptance. Please incorporate the discussed points in your revision.

---

> > > ### Author Response · Authors · 2025-08-01
> > >
> > > We sincerely appreciate your insightful comments and support. We will include all the new discussions and results in our revised paper.

---

### Official Review · Reviewer_WopZ · 2025-07-08

**Clarity:** 2
**Significance:** 2
**Originality:** 2
**Rating:** 3
**Confidence:** 5

**Summary:**

This paper proposes BAM-ICL for hijacking LLMs through ICL. Unlike conventional adversarial attacks, BAM-ICL distributes a fixed perturbation budget across multiple in-context examples to subtly influence the model's output. It operates in two stages: an offline stage, which constructs an optimal perturbation budget profile using a global gradient-based method, and an online stage, which perturbs ICEs sequentially as they arrive. Experiments show that BAM-ICL outperforms existing baselines in attack success, stealthiness, and transferability, posing a practical and stealthy threat to ICL-based LLM usage.

**Questions:**

See weaknesses.

**Ethical Concerns:**

["NO or VERY MINOR ethics concerns only"]

**Limitations:**

No.

**Paper Formatting Concerns:**

No.

**Quality:**

2

**Strengths And Weaknesses:**

Strengths:

1. The BAM-ICL method introduces a well-structured two-stage pipeline, with an offline global budget profiling and an online sequential attack mechanism. This mirrors practical settings where ICEs arrive incrementally, enhancing the realism and applicability of the attack.
2. Unlike prior works that use fixed perturbation budgets across ICEs, BAM-ICL learns to allocate the perturbation budget optimally, improving both effectiveness and stealthiness.
3. The method consistently achieves higher Attack Success Rates (ASR) across various models and datasets compared to strong baselines, showing its superior effectiveness.
4. Adversarial ICEs and learned budget profiles transfer well across different LLM architectures, demonstrating strong generalization and practical impact.
5. The paper is well-organised and easy to follow. The authors have provided detailed figures and algorithms to present the proposed approach.

Weaknesses:

1. While the method is tested across benchmark datasets and model families, the experiments are mostly constrained to synthetic or classification tasks. There is limited exploration of more realistic LLM applications, e.g., multi-turn dialogue, reasoning, or code generation,  where the ICL dynamics might differ.
2. The offline stage assumes the attacker can access data drawn from the same distribution as the target task. This assumption may not always hold in real-world settings, especially in proprietary or closed-domain applications.
3.  The evaluation focuses on one prior defense, e.g., prepending clean ICEs, but does not benchmark against more recent or certified defenses, such as detection-based or filtering-based methods.
4. Although the paper uses cosine similarity and perplexity to measure text quality and stealthiness, human evaluation is not included. This may leave room for unnoticed semantic drift or grammatical degradation.
5. The authors claim that GGI is easily detectable while BAM-ICL is more stealthy. However, beyond perplexity comparisons, they should include experimental results using detection-based or filtering-based defenses and provide examples of attacks that fail.
6. Some tables and figures are unclear. For example, what are the different lines shown in Figure 3? It is better to show the legend. Table 1 is too busy, and it is hard to tell which one achieves the better performance. It is not easy to know that "the loss increases more rapidly than with the attack with a flat budget" as shown in Figure 6.
7. Since GGI is the most related work, the authors should also have conducted comparisons on the cosine similarity distribution histograms of GGI in Figure 5 and the transferability of GGI in Table 2.
8. It is better to show some adversarial examples in the main concept or the appendix to show the stealthiness of the proposed method.

---

> ### Author Rebuttal · Authors · 2025-07-31
>
> We sincerely appreciate your constructive suggestions! Please kindly find our response to each question below. Due to character constraints, we have abbreviated the questions for brevity.
>
> **W1. Exploration of more realistic applications.**
>
> Our initial experiments primarily followed the settings used in prior works on hijacking attacks, which mostly focused on classification tasks. Below, we provide additional experimental results on jailbreaking and reasoning tasks. We will incorporate a more comprehensive evaluation and detailed analysis in the later version.
>
> **Reasoning Tasks**
> We follow the settings of [1] to use SuperGLUE [2] to benchmark the reasoning performance of ICL on OPT-30B with $n=5$. The accuracy of different categories is shown in the table below. It can be seen that BAM-ICL outperforms Flat attack and is close to the Global attack, which exhibits a similar trend as classification tasks presented in the paper.
> |Method|BoolQ|RTE|WIC|WSC|
> |-|-|-|-|-|
> |CA|76.5|52.4|51.1|61.6|
> |+Global|37.4|30.1|29.3|35.3|
> |+Flat|54.6|40.4|40.2|43.1|
> |**+BAM-ICL**|39.6|33.4|40.1|37.9|
>
> **Jailbreaking Tasks**
> We use AdvBench [3] as in GGI [4]. The attack success rate (ASR) on Mistral-7b-instruct is shown below. Since both BAM-ICL and GGI can achieve an ASR of ~99% at $n$=4 based on the original budget of GGI, we conduct this comparison at half of the budget. We also include a Zero-Shot baseline for comparison, which provides the model with only the malicious prompt. Similar to the behavior on classification tasks, as BAM-ICL depends on ICE positioning and the associated budget profile, it benefits from having more ICEs and yields better performance with a larger $n$. It is also important to note that BAM-ICL achieves better perplexity.
>
> |Method|$n$=2|$n$=4|$n$=12|
> |-|-|-|-|
> |Zero-Shot|42.6|42.6|42.6|
> |GGI|70.1|79.7|86.3|
> |**BAM-ICL**|63.7|77.2|**94.9**|
>
> **W2. Assumption of drawing in-distribution data.**
>
> We agree that for some highly sensitive applications, such as medical data, our assumption of drawing in-distribution data may not hold. However, it is worth noting that this assumption, which allows for a slightly stronger attacker with access to task-relevant public data, is widely adopted in AI security research, for instance, adversarial attack [5], backdoor attack [6], membership inference attack [7] as well as recent works on LLM security [8]. This threat model also provides a credible worst-case scenario for the defender. For most real-world settings, the assumption is valid based on the fact that most tasks (e.g., sentiment analysis, question answering, and code generation) are constructed and trained from publicly available datasets (e.g., Wikipedia, GitHub). We acknowledge that exploring attack strategies in settings where the attacker does not have access to in-distribution data represents an important and challenging direction for future work.
>
> **W3. More detection-based or filtering-based defenses.**
>
> In our work, we primarily compared with GGI and therefore adopted the same defense strategies used in that setting for consistency. We would like to emphasize that the core contribution of our paper lies in the proposed budgeted two-phase attack strategy, while the choice of perturbation generation method can potentially incorporate various alternatives. In this regard, BAM-ICL improves stealthiness by strategically distributing perturbations across the context examples, rather than concentrating them on individual inputs. We have conducted additional experiments against several selected defense techniques below. We will include more results in the later version.
>
> **Additional Experiments against Defenses**
>
> One straightforward filtering method is to employ a perplexity-based filter [9] as a defense strategy, which is also used in [10]. As shown in Fig. 2(b) of the main paper, BAM-ICL achieves better perplexity and hence is expected to perform well against such perplexity-based filtering defenses. Besides, we have also conducted additional experiments against three representative yet suitable defenses from different categories: *paraphrasing* [9] that runs in conjuction with a detector to test the performance against paraphrasing suspicious words within single ICE, *In-Context Example Selection* [1] as a detection-based defense across multiple ICEs, and *Fantastically Ordered Prompts* [11] for shuffling defense across multiple ICEs. The results of ASR drop percentage (*lower represents better robustness against defenses*）for OPT-30B on SST-2 are reported.
>
> |Defense strategy|Method|$n$=3|$n$=12|
> |-|-|-|-|
> |**Perplexity&nbsp;Filter&nbsp;(Individual&nbsp;ICE)**|Flat|19.85|26.64|
> ||Global|19.99|27.45|
> ||GGI|31.66|39.83|
> ||**BAM-ICL**|**17.57**|**19.63**|
> |**Paraphrasing&nbsp;(Individual&nbsp;ICE)**|Flat|24.58|24.56|
> ||Global|22.67|24.11|
> ||GGI|34.13|31.02|
> ||**BAM-ICL**|**20.17**|**21.13**|
> |**Detection&nbsp;(Across&nbsp;ICEs)**|Flat|23.68|26.74|
> ||Global|20.53|20.34|
> ||GGI|22.32|**17.66**|
> ||**BAM-ICL**|**19.66**|22.31|
> |**Shuffling&nbsp;(Across&nbsp;ICEs)**|Flat|**16.88**|**14.68**|
> ||Global|34.46|43.79|
> ||GGI|20.57|27.33|
> ||**BAM-ICL**|24.55|27.37|
>
> It can be seen that BAM-ICL demonstrates stronger resilience against filtering and detection-based defenses than GGI, both within individual ICEs and across multiple ICEs. However, shuffling and reordering could impact the effectiveness of BAM-ICL, compared to Flat attack, which aligns with our expectations, since BAM-ICL relies on position-dependent perturbation budget allocation.
>
> **W4. Human evaluation is not included.**
>
> Due to the time limit during the rebuttal, we are not able to complete the approval process and setup required for human evaluation. However, we included examples at the end of our appendix (Section H) for the readers, as human judges, to verify the stealthiness and text quality.
>
> **W5. Experimental results using detection-based or filtering-based defenses and provide examples of attacks that fail.**
>
> Please refer to our responses to W3 regarding additional defense results. Here, we provide an example that GGI fails against *paraphrasing* [9]. Note that GGI appends adversarial suffixes to ICEs. The outputs are highlighted in bold.
>
> **clean ICEs**
>   - Review: confusing plot that makes no sense
>
>     Sentiment: negative
>   - Review: outstanding performances by the entire cast
>
>     Sentiment: positive
>   - Review: compelling narrative with excellent character development
>
>     Sentiment: positive
>   - Review: hide new secretions from the parental units
>
>     Sentiment: **negative**
>
> **BAM-ICL**
>   - Review: confusing *Ploty* that makes no sense
>
>     Sentiment: negative
>   - Review: *Great* performances *via aforementioned* entire cast
>
>     Sentiment: positive
>   - Review: compelling narrative *together with quality*
>
>     Sentiment: positive
>   - Review: hide new secretions from the parental units
>
>     Sentiment: **positive**
>
> **GGI**
>   - Review: confusing plot that makes no sense *NULL (did not pass detector, paraphrased)*
>
>     Sentiment: negative
>   - Review: outstanding performances by the entire cast *Remove*
>
>     Sentiment: positive
>   - Review: compelling narrative with excellent character development *NULL (did not pass detector, paraphrased)*
>
>     Sentiment: positive
>   - Review: hide new secretions from the parental units
>
>     Sentiment: **negative** (fail)
>
> **W6. Some tables and figures are unclear.**
>
> We will follow your suggestion and update tables and figures in the revision to make sure they are clear and easy to understand.
>
> For Figure 3, we illustrate the budget profiles computed under varying total perturbation budgets. We will include legends in the revised version for better clarity. The key message we aim to convey is that the budget profile remains relatively consistent for a given task, highlighting its task-specific nature rather than dependence on a fixed budget value.
>
> For Table 1, we will remove the error stats from the main paper and move those to appendix to improve readability.
>
> For the discussion of Figure 6, we will rephrase it from "increases more rapidly" to "is consistently higher".
>
> **W7. Cosine similarity distribution histograms of GGI in Figure 5 and the transferability of GGI in Table 2.**
>
> Table 2 demonstrates the budget profile transferability of BAM-ICL. However, since the budget profile is unique to our work (in fact, BAM-ICL is the first work to consider the budget allocation in ICL hijacking attack), it is not applicable to evaluate GGI for budget transferability.
>
> We report the cosine similarity distribution based on the examples provided in the GGI paper. The results show that BAM-ICL can achieve higher cosine similarity.
>
> |Method|$n$=2|$n$=4|$n$=8|
> |-|-|-|-|
> |**BAM-ICL**|**0.95 ± 0.05**|**0.92 ± 0.03**|**0.83 ± 0.06**|
> |GGI|0.92 ± 0.09|0.89 ± 0.04|0.71 ± 0.09|
>
> **W8. Attack examples.**
>
> Please kindly see the examples included at the end of our appendix (Section H).
>
> **References**
>
> [1] In-context example selection with influences. arXiv:2302.11042, 2023.
>
> [2] SuperGLUE: A stickier benchmark for general-purpose language understanding systems. NeurIPS, 2019.
>
> [3] Why should adversarial perturbations be imperceptible? Rethink the research paradigm in adversarial NLP. EMNLP, 2022.
>
> [4] Hijacking large language models via adversarial in-context learning. arXiv:2311.09948, 2023.
>
> [5] Transferability in machine learning: From phenomena to black-box attacks using adversarial samples. arXiv:1605.07277, 2016.
>
> [6] Clean-label backdoor attacks on video recognition models. CVPR, 2020.
>
> [7] Membership inference attacks against machine learning models. S&P, 2017.
>
> [8] Improving few-shot performance of language models. ICML, 2021.
>
> [9] Baseline defenses for adversarial attacks against aligned LMs. arXiv:2309.00614, 2023.
>
> [10] AdvPrompter: Fast adaptive adversarial prompting for LLMs. ICML, 2025.
>
> [11] Fantastically ordered prompts and where to find them: Overcoming few-shot prompt order sensitivity. ACL, 2022.

---

> > ### Author Response · Authors · 2025-08-05
> >
> > Thank you once again for your valuable comments. If you have any further questions regarding our response, please kindly let us know. We would be happy to address them promptly during the discussion period.

---

### Note · Authors · 2025-08-12

We sincerely thank all the reviewers for their time, constructive feedback, and insightful suggestions, which have helped us strengthen our work. We are also deeply grateful to the ACs for the consideration. This paper introduces BAM-ICL, the first method--to the best of our knowledge--that exploits in-context learning (ICL) to hijack LLMs with budgeted adversarial manipulation. The main contribution lies in the proposed budgeted two-phase attack strategy for hijacking. Our extensive experiments demonstrate consistent improvements over existing baselines and across classification, reasoning, and jailbreaking tasks. We believe all reviewers' concerns and questions have been addressed in our responses.

In particular, following the reviewers' suggestions, we conducted additional experiments during the rebuttal, summarized as follows:

- **Evaluations on Reasoning and Jailbreaking Tasks** (including jailbreaking results on more robustly aligned models LLaMA-3.1-8B-Instruct and Qwen-2.5-7B-Instruct as suggested during the discussion period): These results further confirm our method's performance advantages and validate the effectiveness of the proposed budgeted strategy.

- **Comparisons against SOTA Defenses**: BAM-ICL demonstrates stronger resilience against filtering and detection-based defenses than GGI, both within individual in-context examples (ICEs) and across multiple ICEs.

- **Results on Time Complexity**: BAM-ICL offers significantly lower time complexity than the Global attack. Even when accounting for the additional cost of the offline phase, the overall runtime increase remains modest.

- **Attack Examples**: In addition to the attack examples included in the appendix, we have provided more attack examples to show the stealthiness and text quality of our method.

- **Cosine Similarity of Attack Examples**: The results show that BAM-ICL can achieve higher cosine similarity.

These results highlight BAM-ICL's novelty, effectiveness, practical relevance, and broad applicability. We have also clarified the questions regarding the threat model and problem setting, as well as the presentation of some tables and figures in our responses. We will include all the new discussions and results in the revised paper.

Overall, we believe BAM-ICL advances the understanding of attack strategies in LLMs, offering new insights into their vulnerabilities and potential defenses. Thanks again for all the time and effort in evaluating our paper.

---

### Decision · Program_Chairs · 2025-09-17

**Decision:**

Accept (poster)

**Comment:**

This paper introduces BAM-ICL, an attack framework that targets Large Language Models under in-context learning (ICL) by dynamically allocating a limited adversarial budget across the demonstration examples. The method operates in two stages: an offline stage, which computes an optimal perturbation budget profile using a global gradient-based strategy; and an online stage, where perturbations are generated progressively in response to incoming examples.

Experimental results show that BAM-ICL consistently achieves higher attack success rates across a range of models and datasets compared to strong baselines, demonstrating its superior effectiveness. The framework’s design, which mirrors the sequential arrival of in-context examples, closely aligns with real-world usage scenarios. However, one reviewer raises a valid concern: the assumption in the offline stage that the attacker has access to data drawn from the same distribution as the target task may not hold in all practical settings.

Overall, this is a timely and well-executed work with clear relevance. It deserves serious consideration for a poster presentation.